# Developmental system drift and modular gene regulatory networks shape gastrulation in *Acropora*

Juan P Ossa-Gómez[1] , Héctor A Rodríguez-Cabal[1,2] , Alejandro Reyes-Bermúdez[3]

**Although gastrulation is a conserved morphogenetic process in animals, the cellular mechanisms underlying it are highly variable. This raises important questions regarding the extent to which conserved or divergent gene regulatory programs (GRNs) control gastrulation in phylogenetically distant taxa. We compared gene expression profiles during gastrulation of *Acropora digitifera* and *Acropora tenuis*, species that diverged ~50 million years ago. Despite the morphological similarity, each species uses divergent GRNs, supporting the concept of developmental system drift. Consistently, orthologous genes showed significant temporal and modular expression divergence, indicating GRN diversification rather than conservation. Yet, we identified a subset of 370 differentially expressed genes that were up-regulated at the gastrula stage in both species, with roles in axis specification, endoderm formation, and neurogenesis, suggesting a conserved regulatory "kernel" for the process. We also identified species-specific differences in paralog usage and alternative splicing patterns that indicate independent peripheral rewiring of this conserved module. Interestingly, although *A. digitifera* exhibits greater paralog divergence, consistent with neofunctionalization, *A. tenuis* shows more redundant expression, suggesting the regulatory robustness of developmental programs in this species.**

## Introduction

Reef-building corals of the genus *Acropora* belong to the phylum Cnidaria, a monophyletic group of diploblastic metazoans widely accepted as the sister group to the bilaterians (Lanna, 2015; Genikhovich & Technau, 2017). Because of its basal position in the phylum as a member of the Anthozoa class and the existence of publicly available genomic resources (Genikhovich & Technau, 2017; Shinzato et al, 2021), the genus has emerged as a model for studying the evolution of mechanisms involved in development, under the hypothesis that genes and other features shared by

corals and higher metazoans are presumably ancestral (Lanna, 2015; Lebedeva et al, 2021). One of the metazoan developmental events that has attracted significant interest in cnidarians is the morphogenetic process of gastrulation, because of its importance in (1) understanding the gene regulatory programs (GRNs) regulating germ layer formation and the establishment of body plans at the base of metazoan (Martindale, 2005; Yasuoka et al, 2016), and (2) characterizing the high variability of gastrulation strategies observed within the phylum (Kraus & Markov, 2017; Technau, 2020). For instance, within the order Scleractinia, it is possible to observe gastrulation by invagination and gastrulation by bending of the flattened blastula, typical variants of the "robust" and "complex" clades, respectively (Kraus & Markov, 2017). Although the mechanisms regulating this morphogenetic transition exhibit low phylogenetic signal (Kraus & Markov, 2017), evidence shows that an ancestral conserved set of genomic regulatory modules coordinates germ layer differentiation and axial determination across metazoans (Technau, 2020; Schauer & Heisenberg, 2021).

The diversification of gastrulation strategies has been linked to lineage-specific adaptations to diverse ecological niches, resulting in distinct developmental programs (Kalinka & Tomancak, 2012). Therefore, it is reasonable to think that the diversification of gastrulation mechanisms that allow animal embryos to gastrulate under a wide range of physical and ecological constraints (Keller et al, 2003) results from natural selection fueled by the diversity of divergent early developmental GRNs. Several lines of evidence have shown that animal development exhibits different degrees of conservation when comparisons are made throughout development between phylogenetically distant organisms (Domazet-Lošo & Tautz, 2010; Kalinka et al, 2010; Marlétaz et al, 2018). Numerous models have been proposed to predict conservation patterns during development; yet, the hourglass model is the most widely accepted (Irie & Kuratani, 2011, 2014; Kalinka & Tomancak, 2012). This model predicts early and late phases of divergence during ontogeny within a phylum, linked by a morphologically conserved period of mid-embryonic development known as the phylotypic period (Kalinka & Tomancak, 2012).

---

[1]Facultad de Ciencias Exactas y Naturales, Grupo Agrobiotecnología. Universidad de Antioquia, Medellín, Colombia   [2]Departamento de Ciencias Agronómicas, Facultad de Ciencias Agrarias, Grupo Fitotecnia Tropical, Universidad Nacional de Colombia, Medellín, Colombia   [3]Facultad de Ciencias Básicas, Programa de Biología, Universidad de la Amazonía, Florencia, Colombia

Correspondence: an.reyes@udla.edu.co

Animal embryos have been shown to exhibit robustness, maintaining the gastrulation process even when changes in the regulation of specific signaling genes during gastrulation are induced (Kirillova et al, 2018; Chuai et al, 2023), which is consistent with the idea of diversification of GRNs during early animal development predicted by the hourglass model. However, gastrulation is likely not only dependent on GRNs; it has been proposed that geometric and mechanical constraints during cell division in early development play essential roles in establishing the basic animal body plan (Edelman et al, 2016; Steventon et al, 2021). This raises important questions about the conserved and divergent aspects of the molecular networks underlying gastrulation, and how the same basal molecular toolbox is readjusted in different organisms to achieve correct germ layer formation and embryonic development, according to species-specific environmental pressures, mechanical constraints, and geometrical factors (Edelman et al, 2016).

In this context, to understand the degree of conservation and diversification of GRNs underlying gastrulation at the base of animal evolution, we compared gene expression profiles obtained by RNA-seq during the early development of two phylogenetically distant *Acropora* species. *Acropora tenuis* and *Acropora digitifera* are two common species in the Indo-Pacific Ocean that diverged ~50 million years ago (Shinzato et al, 2021). We compared gene expression profiles at three stages of early development: the blastula (PC), the gastrula (G), and the early larval stage known as the sphere (S). In both species, early embryogenesis is marked by the formation of a flattened blastula without a blastocoel known as a prawn chip (PC) (Okubo & Motokawa, 2007), which passes through the gastrula (G) and sphere (S) stages before developing into a planula (P), which eventually settles on the substrate and metamorphoses into an adult polyp (A) (Ball et al, 2004) (Fig 1A). The early development of both species occurs in the plankton, so they share similarities in their developmental environment (Morita et al, 2006; Hayward et al, 2011; Reyes-Bermudez et al, 2016). However, they exhibit different spawning times (Fukami et al, 2003), resulting in reproductive isolation (Morita et al, 2006), differ in settling depth preferences (Suzuki et al, 2008), and show morphological differences in the polyp stage (Veron, 1986; Wallace, 1999; Suzuki et al, 2008).

In addition, gene loss, retention, and duplication have been observed between the two lineages (Mao & Satoh, 2019), and there is evidence showing that duplicated genes are expressed during early development (Shinzato et al, 2021), suggesting that species-specific transcripts may contribute to the diversification of developmental GRNs (Nadimpalli et al, 2015). As both species have reference genomes and established gene models (Shinzato et al, 2021), it is possible to conduct comparative transcriptomics studies to understand how species-specific gene duplication events and differential splicing have shaped developmental GRNs during *Acropora* gastrulation. Lineage-specific events, such as the emergence of in-paralogs or new isoforms, can significantly influence the remodeling of ancestral GRNs underlying early embryonic development and lead to divergence in gastrulation strategies. Gene duplication plays a crucial role in the evolution of transcriptional networks and, therefore, can significantly impact phenotypic evolution (Voordeckers et al, 2015). Evidence shows

that paralogs can diverge after a gene duplication event through gain or loss of function. This divergence can occur through mutations to regulatory regions, resulting in changes in expression profiles that enable new interactions with other genes, potentially acquiring new biological functions (Gagnon-Arsenault et al, 2013). These changes eventually lead to alterations in the dynamics of molecular networks and, subsequently, to evolutionary changes (Ovens et al, 2021).

Likewise, alternative splicing (AS) is a mechanism that increases protein diversity by allowing a single gene to generate multiple proteins without requiring significant genomic changes (Singh & Ahi, 2022). This contributes to the high proteomic complexity despite a limited number of genes (Nilsen & Graveley, 2010). Moreover, it has been demonstrated that the presence of different isoforms can significantly impact the phenotypes of organisms and promote morphological innovation (Jacobs & Elmer, 2021; Verta & Jacobs, 2022). AS has also been implicated in expanding molecular networks (Niklas et al, 2015; Singh & Ahi, 2022). Therefore, in-paralogs and isoforms are ideal mechanisms by which species-specific alternative subnetworks are formed during early development and/or contribute to GRN robustness.

Within this framework, this study offers a temporal resolution that captures gene expression dynamics during gastrulation in two phylogenetically distant *Acropora* species. It focuses on identifying divergent and conserved molecular components underlying this morphogenetic process. This work contributes to a clearer understanding of the mechanisms involved in the emergence of species-specific alternative transcriptional networks during early developmental transitions at the base of animal evolution. Moreover, our results demonstrate in silico how modularity and plasticity in coral GRNs have the potential to enable developmental stability alongside enhancing evolutionary innovation. This work positions *Acropora* as a valuable cnidarian model in evo-devo and provides insights into the molecular basis of coral resilience in changing environments.

## Results

### Transcriptome processing and characterization

Our dataset consisted of nine libraries (including triplicates) representing blastula (PC), gastrula (G), and postgastrula (S) from the *A. digitifera* and *A. tenuis* life cycles (Fig 1A). After quality filtering, we obtained ~30.5 and 22.9 million reads for *A. digitifera* and *A. tenuis*, respectively. Filtered reads were aligned against reference genomes (assembly accession: GCA_014634065.1 for *A. digitifera* and GCA_014633955.1 for *A. tenuis*), resulting in 68.1–89.6% and 67.51–73.74% of the reads (per stage) mapped to the *A. digitifera* and *A. tenuis* genomes, respectively (Table 1). Aligned reads were assembled, resulting in 38,110 merged transcripts for *A. digitifera* and 28,284 for *A. tenuis* (Table 1). The difference in transcript number between species may be explained by a greater sequencing depth in *A. digitifera*, which enabled the detection of more low-abundance transcripts, thereby increasing the total number of assembled transcripts reported for the species.

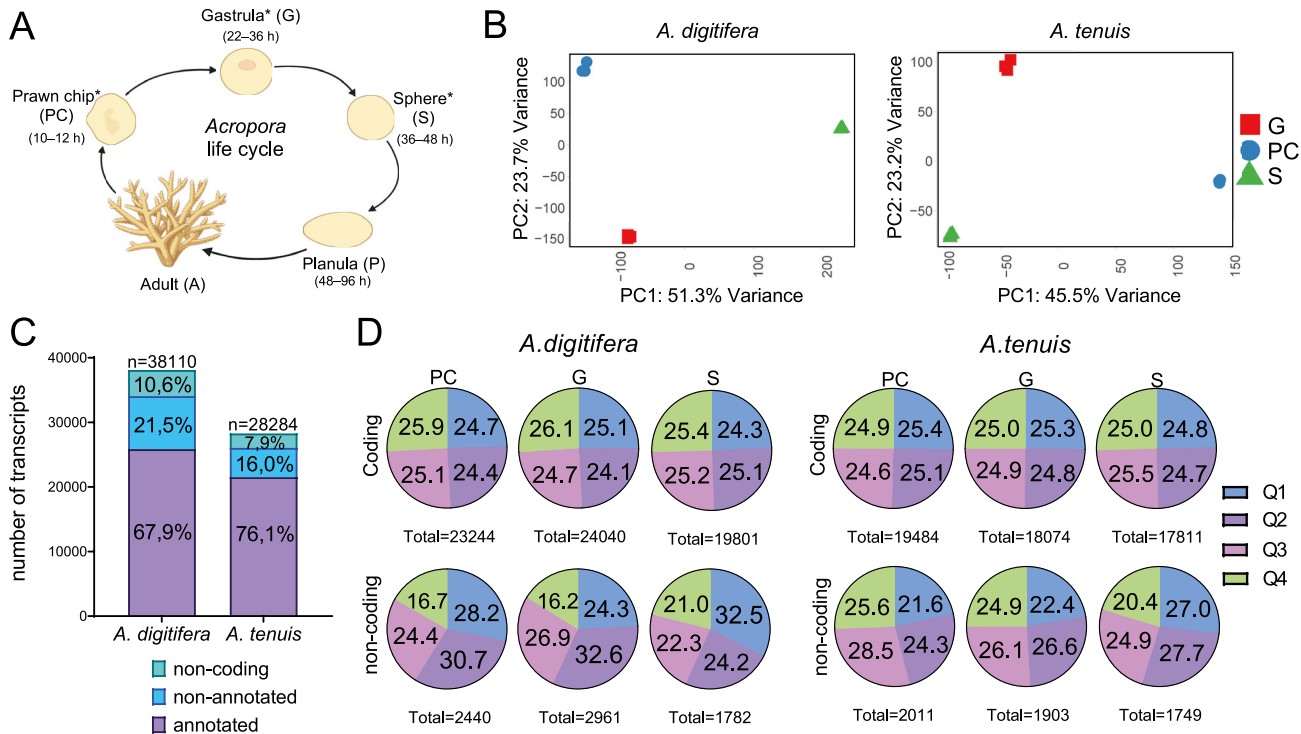

**Figure 1. *Acropora* life cycle and transcriptome characterization.**
**(A)** *Acropora* species are characterized by a flattened blastula without a blastocoel, known as a prawn chip (PC) (10–12 HPF). The blastula thickens, forming a blastopore at the animal pole. Germ layer differentiation (22–36 HPF) indicates the beginning of the gastrula (G) stage. The blastopore closes, creating a mobile spherical embryo (36–48 HPF) called the sphere stage (S). **(A)** Then, larvae elongate along the oral/aboral axis (48–96 HPF) to form a planula (P), which subsequently settles on the substrate to metamorphose into primary polyps that originate new colonies (A). Asterisk (*) shows stages evaluated in this study. **(B)** Principal component analysis of RNA libraries at the PC, G, and S stages of *A. digitifera* and *A. tenuis* validates libraries as replicates. **(C)** Total number of annotated, nonannotated, and noncoding transcripts found in the complete transcriptomes for both species. **(D)** Quartile distribution of the abundance of coding and noncoding transcripts in each life cycle stage for both species.

Although differences in sequencing depth made comparing low-abundant transcripts difficult, we could still compare a large proportion of differentially expressed genes (DEGs) during gastrulation between species. Likewise, sequencing was conducted in 2012, when long-read sequencing technologies were not yet commonly used for RNA sequencing. The use of short-read sequencing technologies in this study may be a methodological limitation affecting assembly quality and downstream analysis. Despite this, principal component analysis (PCA) validated replicates, showing that for *A. digitifera*, transcription profiles in PC and G were closer to each other than to S. In contrast, for *A. tenuis*, the most dissimilar stage was the PC stage (Fig 1B).

BLASTx searches revealed that although 84.10% (32,050) and 82.77% (23,409) of the transcripts were mapped to the *A. digitifera* and *A. tenuis* proteomes, respectively, 4.92% (1,876) and 8.91% (2,520) of the transcripts were mapped to other protein databases, respectively. Likewise, we found that 0.35% (134) and 0.45% (128) of the transcripts were coding transcripts not represented in the publicly available protein datasets, respectively. These subsets of molecules are likely to represent lineage-specific genes. The remaining 10.63% (*A. digitifera*) and 7.87% (*A. tenuis*) (4,050 and 2,227, respectively) were identified as noncoding transcripts. In general, 89.37% and 92.13% of the transcripts were identified as coding for *A. digitifera* and *A. tenuis*, respectively. From these,

25,858 and 21,516 transcripts (67.9 and 76.1%, respectively) were annotated sequences, and the remaining 8,202 and 4,541 transcripts (21.5% and 16.0%, respectively) were not annotated (Fig 1C).

Normalized expression distribution (transcripts per million, TPM) showed that coding sequences in *A. digitifera* were significantly more abundant in the upper ranges of the distribution (Q4 and/or Q3) at the PC and G stages ($P < 0.01$, chi-square goodness-of-fit test), indicating a tendency toward high coding transcript abundance. In contrast, no significant differences among quartiles were observed in other states, including all three stages assessed in *A. tenuis* ($P > 0.05$), suggesting a uniform distribution, that is, a balanced proportion of genes with low, medium, and high expression levels. On the other hand, in *A. digitifera*, noncoding transcripts were more abundant in the lower ranges (Q1 and Q2) in PC and S, and in the intermediate ranges (Q2 and Q3) in G. In contrast, in *A. tenuis*, they tended to be more abundant in the upper ranges of distribution in PC, in the intermediate ranges in G, and in the lower ranges in S (Fig 1D). In all cases, the differences in noncoding transcript distribution across quartiles were statistically significant ($P < 0.05$). Although transcripts in the higher ranges of the distribution are likely to regulate conserved processes occurring globally in developing embryos, low-abundant transcripts might be expressed by specific cell populations. To test this idea, in situ hybridizations

**Table 1. Statistics of RNA-seq reads and assembled transcripts per replicate and stage for *A. digitifera* and *A. tenuis*.**

| Stage | Sample | Number of reads | | Total readings per stage | | Alignment rate | | General alignment rate | | Merged transcripts | | Merged total transcripts | |
|---|---|---|---|---|---|---|---|---|---|---|---|---|---|
| | | *A. digitifera* | *A. tenuis* | *A. digitifera* | *A. tenuis* | *A. digitifera* | *A. tenuis* | *A. digitifera* | *A. tenuis* | *A. digitifera* | *A. tenuis* | *A. digitifera* | *A. tenuis* |
| PC | PC1 | 2,331,358 | 3,022,820 | | | 89.16% | 67.16% | | | | | | |
| | PC2 | 4,145,804 | 2,388,494 | 9,953,579 | 8,007,380 | 89.43% | 67.62% | 89.30% | 67.51% | 27,702 | 18,848 | | |
| | PC3 | 3,476,417 | 2,596,066 | | | 89.24% | 67.80% | | | | | | |
| G | G1 | 4,029,510 | 2,890,575 | | | 89.66% | 70.56% | | | | | 38,110 | 28,284 |
| | G2 | 4,136,915 | 2,461,253 | 13,777,953 | 8,003,807 | 89.54% | 58.85% | 89.61% | 68.67% | 29,024 | 15,164 | | |
| | G3 | 5,611,528 | 2,651,979 | | | 89.63% | 71.17% | | | | | | |
| S | S1 | 2,179,043 | 1,609,827 | | | 68.02% | 74.56% | | | | | | |
| | S2 | 2,382,031 | 2,188,956 | 6,804,233 | 6,873,883 | 68.20% | 74.40% | 68.08% | 73.74% | 21,836 | 16,419 | | |
| | S3 | 2,243,159 | 3,075,100 | | | 68.02% | 72.82% | | | | | | |

and scRNA-seq are necessary, as the spatial patterns of gene expression were out of the scope of this study because of technical and financial limitations.

## Different instructions, same outcome: *Acropora* species use distinct transcriptional programs at morphologically similar stages during gastrulation

To characterize and compare gene expression in *Acropora* during gastrulation, DEGs were identified between consecutive stages (PC versus G and G versus S) for each species. A total of 18,497 DEGs were identified for *A. digitifera*, and 9,486 were identified for *A. tenuis*. In *A. digitifera*, 11,333 DEGs were found in PC versus G and 13,406 in G versus S (Fig 2A). In contrast, in *A. tenuis*, 6,914 DEGs were identified in PC versus G, and 4,325 were identified in G versus S (Fig 2B). For *A. digitifera*, DEGs were more abundant in the G-to-S progression, and for *A. tenuis*, in the PC-to-G progression (Fig S1A). In both species, noncoding transcripts accounted for ~10% of the DEGs in all comparisons (Fig S1B). Table S1 summarizes DEG annotation, including several TF families, such as Sox, Fox, Hes, and Pax, and components of the Wnt, BMP, FGF, and Notch signaling pathways.

Functional enrichment analysis of DEGs revealed an overrepresentation of various biological processes (BPs) between species during gastrulation. In *A. digitifera*, GTPase signaling (GO:0007264) and DNA replication (GO:0034725) were overrepresented in PC, and axis specification (GO:0009798, GO:0009950) and Wnt signaling (GO:0060071, GO:0016055) were overrepresented in G in the PC-versus-G transition. In *A. tenuis*, the BPs enriched in the PC-versus-G transition were extracellular matrix organization (GO:0030198) and cell differentiation (GO:0030855) in PC, and pattern specification (GO:0007389) and ciliary and flagellar motility (GO:0001539) in G. Likewise, in the subsequent G-versus-S transition, in *A. digitifera*, chromosome organization (GO:0051276) was enriched in G, and cilium movement (GO:0003341, GO:0001539) and anatomical structure morphogenesis (GO:0048646) were enriched in S. In *A. tenuis*, Wnt (GO:0060070) and BMP signaling (GO:0030509) were overrepresented in G, and gene expression (GO:0010467) and the cell cycle (GO:0022403) were overrepresented in S (Table 2).

K-means clustering of DEG expression patterns revealed six coexpression clusters in both species (C1 to C6) (Fig 2C and D). Although C1-grouped DEGs up-regulated in PC, C2-grouped DEGs up-regulated in PC and G. C3 DEGs up-regulated only in G, C4 DEGs up-regulated in G and S, and C5 DEGs only up-regulated in S. C6-grouped DEGs up-regulated in PC and S. Despite this, the number of DEGs in each cluster varied between species. Although the most populated cluster for *A. digitifera* was AdC2, containing 23.47% (4,476) of all DEGs for the species (19,069) (Fig 2C), the most abundant cluster for *A. tenuis* was AtC1, grouping 34.40% (3,264) of all DEGs (9,486) (Fig 2D). Interestingly, neuronal fate specification (GO:0048665) was enriched (FDR < 0.05) in cluster C4 for both species, indicating activation of this process at G and S in the two species (Fig 2C and D), and although AdC5 was enriched with molecules with roles in cilium movement and axial specification, AtC5 did so with molecules with roles in cell cycle. Cluster AdC6 did not show any BP-enriched category. BP enrichment for each cluster is summarized in Fig 2C and D.

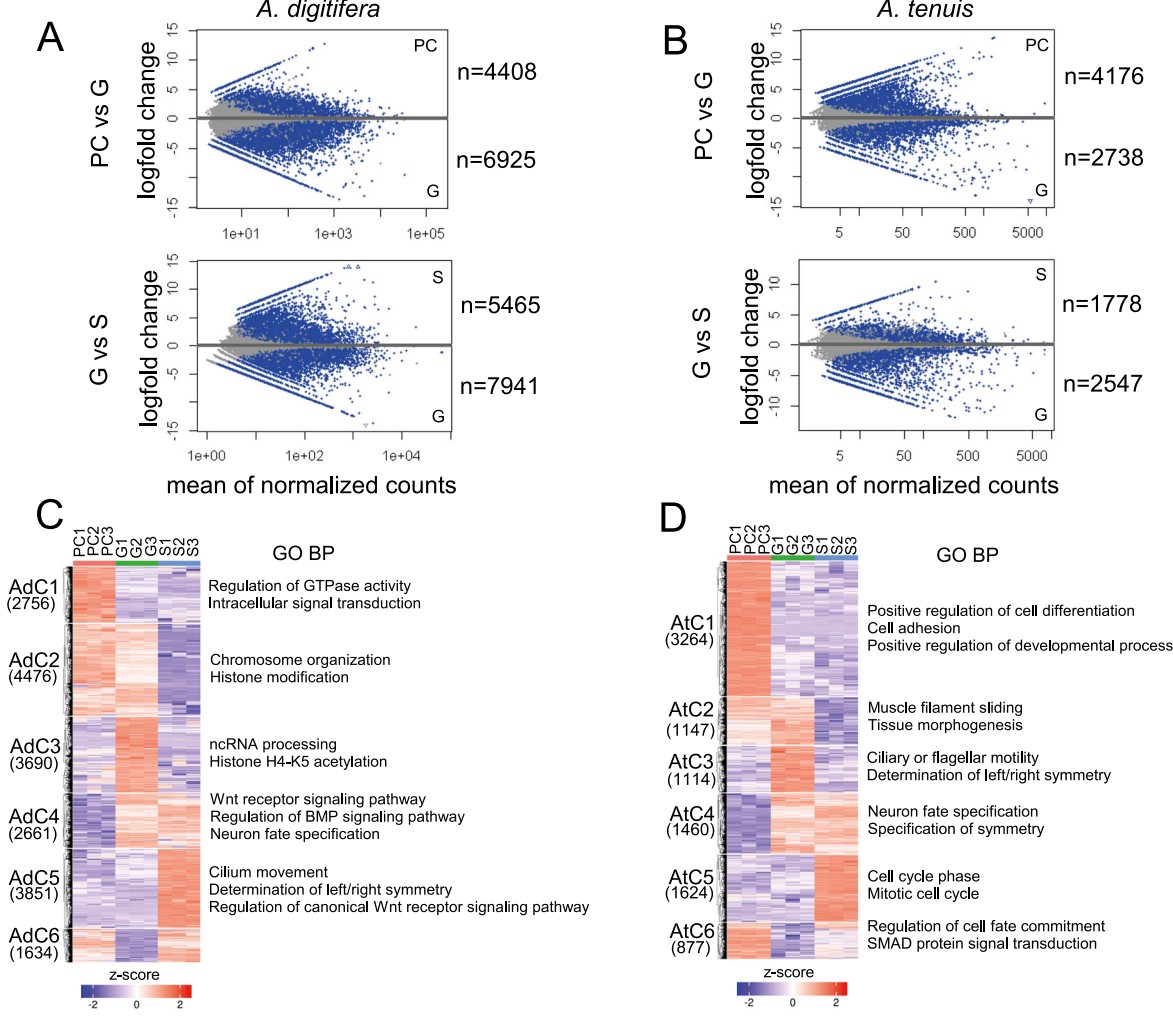

**Figure 2.   Differential expression analysis during gastrulation of *A. digitifera* and *A. tenuis*.**
**(A, B)** MA plot of significantly expressed transcripts (*Padj* < 0.05) in each comparison for *A. digitifera* (A) and *A. tenuis* (B). Significantly expressed transcripts are highlighted in blue. The number of differentially expressed genes (*Padj* < 0.05 and LFC ≥ 1) is indicated for each stage. **(C, D)** K-means clustering of differentially expressed genes identified six expression clusters (C1–C6) for both *A. digitifera* (C) and *A. tenuis* (D). Clusters with no associated BP terms indicate no enrichment for a particular category within the group. Significantly enriched gene ontology (GO) terms (FDR < 0.05) are shown for each cluster.
Source data are available for this figure.

## Divergent programs, similar components: species-specific transcriptional modules contain differentially expressed orthologs that map or regulate developmental signaling pathways

To characterize conserved regulatory gene networks up-regulated during gastrulation, we identified orthologs between the proteomes of the two *Acropora* species. We compared the transcription profiles of "one-to-one" orthologous pairs. We found that 8,245 *A. digitifera* and 7,359 *A. tenuis* molecules exhibited orthologous relationships, represented by 10,308 orthologous pairs (Table S2), organized into 6,896 orthologous groups. From these, "one-to-one" relationships accounted for 86% of all orthologs, "one-to-many" for 12%, and "many-to-many" for ~2%. "One-to-many" relationships were further subdivided into "one-to-many-*A. digitifera*" (one *A. tenuis* ortholog to many *A. digitifera*),

accounting for ~10% of all relationships, and "one-to-many-*A. tenuis*" (one *A. digitifera* ortholog to many *A. tenuis*), accounting for ~2% of all orthologs (Fig 3A). The high proportion of one-to-one orthologs highlights the strong genomic conservation between these two coral species. Conversely, the presence of a higher frequency of one-to-many relationships in *A. digitifera* may reflect species-specific gene duplication events, which could contribute to adaptive traits.

To identify the usage of conserved transcriptional modules during gastrulation in *Acropora*, we focused on 5,965 one-to-one orthologous pairs (86.5% of all orthologs). From these, a subset of 1,629 orthologs were differentially expressed in both species (Fig 3B; Table S3). Differentially expressed orthologs included conserved TF families associated with developmental processes and some components of the Wnt, FGF, BMP, and Notch signaling pathways (Table 3). Interestingly, this module was enriched in BPs

**Table 2.** GO enrichment analysis (BP) of the differentially expressed genes identified in the "PC-versus-G" and "G-versus-PC" comparisons in *A. digitifera* (Ad) and *A. tenuis* (At).

| Comparison | Stage | GO-ID | corr.pvalue | x | n | Description |
|---|---|---|---|---|---|---|
| AdPC versus AdG | AdPC | 7165 | $5.02 \times 10^{-3}$ | 569 | 4,034 | Signal transduction |
| | | 43087 | $7.75 \times 10^{-3}$ | 75 | 378 | Regulation of GTPase activity |
| | | 7265 | $1.95 \times 10^{-2}$ | 61 | 307 | Ras protein signal transduction |
| | | 7264 | $3.23 \times 10^{-2}$ | 101 | 591 | Small GTPase-mediated signal transduction |
| | | 34773 | $3.23 \times 10^{-2}$ | 6 | 9 | Histone H4-K20 trimethylation |
| | | 35407 | $3.45 \times 10^{-2}$ | 4 | 4 | Histone H3-T11 phosphorylation |
| | | 70 | $4.36 \times 10^{-2}$ | 60 | 319 | Mitotic sister chromatid segregation |
| | | 34723 | $4.45 \times 10^{-2}$ | 14 | 43 | DNA replication–dependent nucleosome organization |
| | | 32488 | $2.43 \times 10^{-2}$ | 7 | 11 | Cdc42 protein signal transduction |
| AdPC versus AdG | AdG | 60071 | $6.12 \times 10^{-11}$ | 68 | 170 | Wnt receptor signaling pathway, planar cell polarity pathway |
| | | 16055 | $1.01 \times 10^{-6}$ | 140 | 532 | Wnt receptor signaling pathway |
| | | 1510 | $1.08 \times 10^{-4}$ | 45 | 135 | RNA methylation |
| | | 6333 | $8.07 \times 10^{-4}$ | 72 | 268 | Chromatin assembly or disassembly |
| | | 1709 | $7.42 \times 10^{-3}$ | 68 | 270 | Cell fate determination |
| | | 48863 | $8.47 \times 10^{-3}$ | 90 | 380 | Stem cell differentiation |
| | | 9950 | $1.55 \times 10^{-2}$ | 43 | 158 | Dorsal/ventral axis specification |
| | | 10990 | $1.63 \times 10^{-2}$ | 8 | 14 | Regulation of SMAD protein complex assembly |
| | | 45995 | $2.20 \times 10^{-2}$ | 81 | 348 | Regulation of embryonic development |
| | | 9,798 | $3.59 \times 10^{-2}$ | 95 | 428 | Axis specification |
| | | 48864 | $3.86 \times 10^{-2}$ | 68 | 291 | Stem cell development |
| | | 165 | $4.43 \times 10^{-2}$ | 115 | 537 | MAPKKK cascade |
| AdG versus AdS | AdG | 35404 | $1.67 \times 10^{-4}$ | 13 | 17 | Histone/serine phosphorylation |
| | | 51276 | $1.70 \times 10^{-4}$ | 472 | 1,967 | Chromosome organization |
| | | 34729 | $1.95 \times 10^{-3}$ | 10 | 13 | Histone H3-K79 methylation |
| | | 6342 | $8.14 \times 10^{-3}$ | 81 | 277 | Chromatin silencing |
| | | 10629 | $9.70 \times 10^{-3}$ | 377 | 1,612 | Negative regulation of gene expression |
| | | 16571 | $9.70 \times 10^{-3}$ | 46 | 138 | Histone methylation |
| | | 16570 | $1.25 \times 10^{-2}$ | 158 | 614 | Histone modification |
| AdG versus AdS | AdS | 3341 | $9.49 \times 10^{-11}$ | 87 | 242 | Cilium movement |
| | | 1539 | $5.96 \times 10^{-7}$ | 53 | 143 | Ciliary or flagellar motility |
| | | 30049 | $1.75 \times 10^{-4}$ | 18 | 33 | Muscle filament sliding |
| | | 7368 | $2.29 \times 10^{-3}$ | 92 | 379 | Determination of left/right symmetry |
| | | 90131 | $2.56 \times 10^{-3}$ | 10 | 15 | Mesenchyme migration |
| | | 50954 | $2.78 \times 10^{-3}$ | 96 | 403 | Sensory perception of mechanical stimulus |
| | | 48646 | $3.82 \times 10^{-3}$ | 452 | 2,428 | Anatomical structure formation involved in morphogenesis |
| | | 9799 | $6.30 \times 10^{-3}$ | 96 | 417 | Specification of symmetry |
| | | 7584 | $9.79 \times 10^{-3}$ | 109 | 493 | Response to nutrient |
| | | 60485 | $3.23 \times 10^{-2}$ | 78 | 350 | Mesenchyme development |
| | | 48665 | $4.72 \times 10^{-2}$ | 24 | 82 | Neuron fate specification |
| AtPC versus AtG | AtPC | 30198 | $5.36 \times 10^{-16}$ | 80 | 313 | Extracellular matrix organization |
| | | 30855 | $3.29 \times 10^{-13}$ | 93 | 451 | Epithelial cell differentiation |

  https://doi.org/10.26508/lsa.202503293   vol 8 | no 11 | e202503293   

**Table 2.  Continued**

| Comparison | Stage | GO-ID | corr.pvalue | x | n | Description |
|---|---|---|---|---|---|---|
| | | 45597 | $4.15 \times 10^{-13}$ | 108 | 568 | Positive regulation of cell differentiation |
| | | 51094 | $1.67 \times 10^{-11}$ | 134 | 818 | Positive regulation of developmental process |
| | | 60429 | $1.27 \times 10^{-9}$ | 161 | 1,120 | Epithelium development |
| | | 7498 | $2.36 \times 10^{-9}$ | 58 | 264 | Mesoderm development |
| | | 16477 | $1.33 \times 10^{-8}$ | 131 | 884 | Cell migration |
| | | 45595 | $3.02 \times 10^{-7}$ | 184 | 1,442 | Regulation of cell differentiation |
| | | 71559 | $1.56 \times 10^{-5}$ | 17 | 50 | Response to transforming growth factor beta stimulus |
| | | 1707 | $3.32 \times 10^{-5}$ | 31 | 143 | Mesoderm formation |
| | | 42573 | $4.66 \times 10^{-5}$ | 11 | 24 | Retinoic acid metabolic process |
| | | 7398 | $1.85 \times 10^{-4}$ | 62 | 414 | Ectoderm development |
| AtPC versus AtG | AtG | 1539 | $4.09 \times 10^{-22}$ | 60 | 170 | Ciliary or flagellar motility |
| | | 7368 | $1.85 \times 10^{-8}$ | 62 | 335 | Determination of left/right symmetry |
| | | 9799 | $2.01 \times 10^{-8}$ | 66 | 371 | Specification of symmetry |
| | | 50954 | $6.23 \times 10^{-6}$ | 51 | 298 | Sensory perception of mechanical stimulus |
| | | 1708 | $3.15 \times 10^{-5}$ | 38 | 204 | Cell fate specification |
| | | 48665 | $4.58 \times 10^{-5}$ | 21 | 79 | Neuron fate specification |
| | | 7389 | $5.94 \times 10^{-5}$ | 124 | 1,062 | Pattern specification process |
| | | 16360 | $2.31 \times 10^{-4}$ | 12 | 32 | Sensory organ precursor cell fate determination |
| | | 14016 | $3.09 \times 10^{-4}$ | 14 | 44 | Neuroblast differentiation |
| | | 71697 | $1.70 \times 10^{-3}$ | 17 | 72 | Ectodermal placode morphogenesis |
| | | 60795 | $4.50 \times 10^{-3}$ | 17 | 78 | Cell fate commitment involved in the formation of primary germ layers |
| | | 30510 | $1.66 \times 10^{-2}$ | 20 | 114 | Regulation of BMP signaling pathway |
| | | 60070 | $1.96 \times 10^{-2}$ | 20 | 116 | Canonical Wnt receptor signaling pathway |
| | | 48880 | $2.04 \times 10^{-2}$ | 18 | 100 | Sensory system development |
| | | 1714 | $2.08 \times 10^{-2}$ | 8 | 26 | Endodermal cell fate specification |
| AtG versus AtS | AtG | 30049 | $1.61 \times 10^{-8}$ | 17 | 35 | Muscle filament sliding |
| | | 7368 | $4.47 \times 10^{-7}$ | 54 | 335 | Determination of left/right symmetry |
| | | 9612 | $6.90 \times 10^{-7}$ | 46 | 266 | Response to mechanical stimulus |
| | | 60537 | $1.35 \times 10^{-6}$ | 66 | 470 | Muscle tissue development |
| | | 48859 | $3.03 \times 10^{-5}$ | 18 | 67 | Formation of anatomical boundary |
| | | 19226 | $3.40 \times 10^{-5}$ | 94 | 836 | Transmission of nerve impulse |
| | | 42692 | $3.54 \times 10^{-5}$ | 67 | 532 | Muscle cell differentiation |
| | | 90131 | $1.34 \times 10^{-4}$ | 8 | 15 | Mesenchyme migration |
| | | 16337 | $1.47 \times 10^{-4}$ | 43 | 301 | Cell–cell adhesion |
| | | 60485 | $2.22 \times 10^{-4}$ | 40 | 277 | Mesenchyme development |
| | | 7389 | $6.31 \times 10^{-4}$ | 106 | 1,062 | Pattern specification process |
| | | 30509 | $9.22 \times 10^{-4}$ | 16 | 74 | BMP signaling pathway |
| | | 48665 | $1.88 \times 10^{-3}$ | 16 | 79 | Neuron fate specification |
| | | 60070 | $2.49 \times 10^{-3}$ | 20 | 116 | Canonical Wnt receptor signaling pathway |
| | | 9952 | $7.86 \times 10^{-3}$ | 49 | 446 | Anterior/posterior pattern formation |
| | | 48318 | $1.65 \times 10^{-2}$ | 6 | 19 | Axial mesoderm development |
| | | 42990 | $2.19 \times 10^{-2}$ | 17 | 115 | Regulation of transcription factor import into nucleus |

**Table 2. Continued**

| Comparison | Stage | GO-ID | corr.pvalue | x | n | Description |
|---|---|---|---|---|---|---|
| | | 14031 | $2.21 \times 10^{-2}$ | 28 | 231 | Mesenchymal cell development |
| | | 50673 | $2.35 \times 10^{-2}$ | 10 | 51 | Epithelial cell proliferation |
| | | 48568 | $2.40 \times 10^{-2}$ | 65 | 675 | Embryonic organ development |
| | | 70371 | $3.59 \times 10^{-2}$ | 6 | 23 | ERK1 and ERK2 cascade |
| AtG versus AtS | AtS | 10467 | $2.84 \times 10^{-16}$ | 320 | 2,644 | Gene expression |
| | | 22403 | $7.08 \times 10^{-15}$ | 178 | 1,217 | Cell cycle phase |
| | | 16458 | $4.60 \times 10^{-7}$ | 62 | 364 | Gene silencing |
| | | 10608 | $1.01 \times 10^{-5}$ | 117 | 931 | Posttranscriptional regulation of gene expression |
| | | 30261 | $5.61 \times 10^{-5}$ | 21 | 82 | Chromosome condensation |
| | | 9988 | $3.10 \times 10^{-3}$ | 15 | 65 | Cell–cell recognition |
| | | 1714 | $4.17 \times 10^{-2}$ | 7 | 26 | Endodermal cell fate specification |
| | | 6323 | $5.53 \times 10^{-5}$ | 40 | 226 | DNA packaging |

x, number of query genes associated with the GO term; n, number of genes in the reference set annotated with the GO term.

related to neuronal fate specification (GO:0045665, GO:0048663, GO:0048665), axial pattern formation (GO:0009798, GO:0009948, GO:0009950), gastrulation (GO:0007369, GO:0060795), Wnt (GO:0060070), BMP (GO:0030509), and Notch (GO:0045747) pathways (Fig 3B; Table S4). Finally, analysis of the distribution of the 1,629 shared orthologs across developmental stages revealed a peak of expression in *A. digitifera* during the G-versus-S transition. In contrast, in *A. tenuis*, the peak occurred during the PC-versus-G transition, suggesting a temporal shift in the activation timing of developmental transcriptional modules between the two species (Fig 3C). These findings not only reveal the existence of a deeply conserved transcriptional module active during gastrulation in both *Acropora* species but also reveal species-specific differences in their temporal regulation.

## Together but not blended: species-restricted expressed orthologs are more abundant in the G-to-S progression in *A. digitifera* and in the PC-to-G transition in *A. tenuis*

In some cases, only one of the orthologous pairs was differentially expressed in one of the two species. This was observed in 2,136 *A. digitifera* and 722 *A. tenuis* orthologs (Table S5). Furthermore, although the subset of orthologs differentially expressed only in *A. digitifera* was enriched for genes related to Wnt (GO:0016055) and EGF (GO:0007173) receptor signaling, axis specification (GO:0009798), DNA damage response (GO:0042770), and morphogenesis (GO:0048856), the subset of orthologs differentially expressed only in *A. tenuis* was enriched with molecules with roles in the organization of myosin II filaments (GO:0031038) (Fig 3B; Table S6) (Fig 3C).

Likewise, when we compared the distribution of orthologous pairs that were differentially expressed in both species for the PC-versus-G comparison, we found that 58% (131 orthologs in PC and 229 in G, totaling 360 orthologs) were differentially expressed at the same stage. The remaining 42% (256 orthologs) showed asynchronous differential expression. In the G-versus-S comparison, 16% of the orthologs (50 in G and 56 in S, totaling 106 orthologs)

were differentially expressed at the same stage, whereas the remaining 82% (573 orthologs) showed asynchronous differential expression (Fig S1C). Interestingly, both AdS and AtS were found to share most of their DEGs with AtG and AdG, respectively. In other words, most of the DEGs found in the S stage of both species were also differentially expressed in the G stage of the opposite species (Fig 3D). Despite this, we identified a subset of 370 orthologs that were up-regulated at the G stage in both species (Table S7). This module included *Bmp1-like*, *Six3-like*, *Prdm14-like*, *Wnt-2b-like*, *Pitx2-like*, and *Sox-2-like*. It was enriched in BPs such as neuron fate specification (GO:0048665), canonical Wnt signaling pathway (GO:0060070), and endodermal cell fate specification (GO:0001714), and is likely to represent the ancestral gastrulation core GRN present in the last common *Acropora* ancestor (Table S8). More research is necessary to test this idea.

## Low transcriptional conservation leads to morphological convergence through the differential use of distinct gene modules

To characterize conserved coexpression modules during gastrulation in *Acropora*, we compared one-to-one orthologous transcription profiles between K-means clusters (Figs 3E and S1D). Overall, we found low similarity in expression patterns between species. From 1,629 differentially expressed orthologous pairs, only 215 (13.2%) exhibited similar expression patterns. From this, 75 DEGs were coexpressed in C1, 9 in C2, 12 in C3, 82 in C4, 28 in C5, and 9 in C6 (Fig S1E). These molecules included *Runx3-like*, *Dvr1-like*, *Barhl1-like*, and *Smad4-like* in C1; *Bmp1-like* in C2; *Sox-14-like*, *Foxj1b-like*, *Dmrta2-like*, *Pitx2-like*, *Tbx20-like*, and *Sox-2-like* in C4; and *Chrdl1-like* and *Sox-14-like* in C5 (Table 4). Only C1 was found to have significant enrichment in BPs such as BMP signaling pathway (GO:0030509) and stem cell differentiation (GO:0048863) (Table S9).

Despite this, our results also identified subsets of orthologous pairs coexpressed in both species, but in different clusters (Table 4). For example, AdC3 and AtC4 (46 orthologs) coexpressed genes such as *Dmrt3-like*, *Arx-like*, *Prdm14-like*, and *Sox-14-like*.

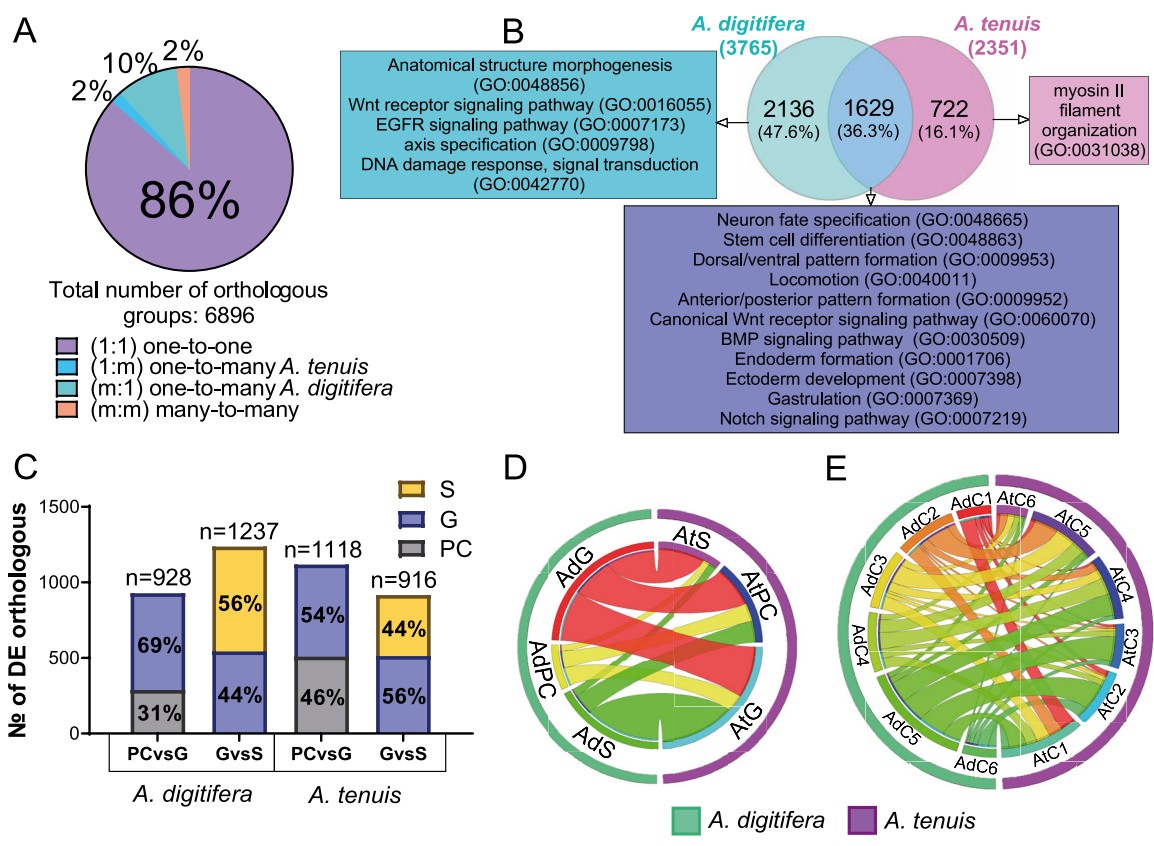

**Figure 3. Differential expression of orthologous genes during gastrulation.**
**(A)** Total number of orthologs from complete transcriptome comparisons, classified into one to one (1:1), one *A. digitifera* to many *A. tenuis*, one *A. tenuis* to many *A. digitifera*, or many to many. **(B)** Differentially expressed one-to-one orthologs, showing both shared and exclusive differentially expressed genes (DEGs) between species. **(C)** Number of shared DEGs by each stage per comparison. **(D)** Overlapping one-to-one DEG orthologs per stage between species. **(E)** Overlapping one-to-one DEG orthologs per K-means clusters between species.

AdC4 and AtC1 (52 orthologs) coexpressed genes such as *Nkx2-like*, *Jag1-like*, *Fgfr1-like*, and *Otx-like*, and were enriched in BPs such as neuron fate determination (GO:0048664). AdC4 and AtC5 (63 orthologs) included genes such as *Hes-1-like*, *Sox-11-like*, and *Dlx4a-like*, and were enriched in the maturation of 5.8S rRNA (GO:0000460). AdC5 and AtC1 (87 orthologs) included genes such as *Nkx2.5-like*, *Sox-21-A-like*, *MafF-like*, *Wnt-1-like*, *FoxA2-A-like*, *Fgfr2-like*, *Fgf18-like*, and *Fgfr3-like* and were enriched in genes related to the positive regulation of neurogenesis (GO:0050769), cellular response to retinoic acid (GO:0071300), and endoderm development (GO:0007492). AdC5 and AtC2 (130 orthologs) coexpressed genes such as *FoxC1-B-like*, *Gsx1-like*, and *Tll1-like genes*. AdC5 and AtC3 (121 orthologs) coexpressed genes such as *Fzd5-like*, *Rx1-like*, *Wnt-7 b-like*, and *Lhx1-like*, and were enriched in BPs such as ciliary or flagellar motility (GO:0001539) and specification of symmetry (GO:0009799). Finally, the AdC5 and AtC4 clusters (105 orthologs) included genes such as *FoxG1-like*, *Six3-like*, *Sox-15-like*, *POU4F3-like*, and *Wnt-2b-like*, and were enriched in cilium movement (GO:0003341). Table S9 summarizes the GO enrichments of the one-to-one orthologous genes shared between groups.

In addition, we focused on the set of shared transcripts found in clusters C2, C3, and C4, which showed increased expression at the G stage. In these clusters, we identified 272 orthologs (Table S10),

including genes such as *Fgfr2-like*, *Notch1-like*, *Bmp1-like*, *Dmrt3-like*, *Dmrta2-like*, *Foxj1b-like*, *Phox2b-like*, *Pitx2-like*, *Arx-like*, *Prdm14-like*, *Sox-14-like*, *Sox-2-like*, *Tbx20-like*, *Tbx3-like*, *Tef-like*, and *Zic4-like*. This set of transcripts was enriched in BPs related to the development of anatomical structures (GO:0030324, GO:0048565), neuron fate specification (GO:0048665), embryonic pattern specification (GO:0009880), and endoderm formation (GO:0001706) (Table S11).

### Paralogous dynamics reveal species-specific strategies during gastrulation: conserved coexpression in *A. tenuis* versus divergent regulation in *A. digitifera*

We identified 10,922 *A. digitifera* and 2,220 *A. tenuis* paralogous pairs expressed during gastrulation (Table S12). Paralogous pairs were classified into P1: both DEGs; P2: one DEG; and P3: no DEG. We identified 1,698 paralogous pairs for *A. digitifera* in the PC-versus-G transition belonging to class P1, 3,753 to class P2, and 5,471 to class P3. During the G-versus-S progression, we found 1,949 paralogous pairs belonging to P1, 4,257 to P2, and 4,716 to P3. Likewise, for *A. tenuis* in the PC-versus-G transition, we identified 1,479, 413, and 328 paralogous pairs belonging to P1, P2, and P3, respectively. In the G-versus-S progression, we found 119 paralogous pairs in P1,

**Table 3. Summary of differentially expressed orthologous gene pairs during the development of *A. digitifera* and *A. tenuis*.**

| Transcript ID | | Description | A. digitifera | | A. tenuis | |
|---|---|---|---|---|---|---|
| A. digitifera | A. tenuis | | PC versus G | G versus S | PC versus G | G versus S |
| adi_MSTRG.16544.1 | ate_MSTRG.11573.1 | Aristaless-related homeobox protein-like | G | G | G | — |
| adi_MSTRG.21148.1 | ate_MSTRG.9278.1 | barH-like 1 homeobox protein | PC | — | PC | — |
| adi_MSTRG.25711.2 | ate_MSTRG.26313.1 | Bone morphogenetic protein 1-like | — | G | — | G |
| adi_MSTRG.15884.1 | ate_MSTRG.1331.1 | Bone morphogenetic protein 1-like | PC | — | G | — |
| adi_MSTRG.16108.1 | ate_MSTRG.25398.1 | Catenin delta-2-like | — | S | G | — |
| adi_MSTRG.17330.2 | ate_MSTRG.15266.1 | Chordin-like | — | S | — | S |
| adi_MSTRG.6785.1 | ate_MSTRG.2867.2 | Cyclic AMP-dependent transcription factor ATF-5-like | — | G | PC | — |
| adi_MSTRG.15320.1 | ate_MSTRG.18248.1 | Diencephalon/mesencephalon homeobox protein 1-B-like | PC | — | — | S |
| adi_MSTRG.386.1 | ate_MSTRG.10325.1 | Doublesex and mab-3 related transcription factor 3-like | G | G | G | — |
| adi_MSTRG.8385.1 | ate_MSTRG.22870.1 | Doublesex- and mab-3-related transcription factor 3a-like | — | G | — | S |
| adi_MSTRG.13695.1 | ate_MSTRG.18015.1 | Doublesex- and mab-3-related transcription factor A2-like | G | G | G | — |
| adi_MSTRG.8075.1 | ate_MSTRG.9709.1 | Fibroblast growth factor 18-like | — | S | PC | — |
| adi_MSTRG.4035.1 | ate_MSTRG.27096.1 | Fibroblast growth factor receptor 2-like | — | S | PC | — |
| adi_MSTRG.8071.1 | ate_MSTRG.5924.1 | Fibroblast growth factor receptor 2-like | — | S | PC | — |
| adi_MSTRG.4038.1 | ate_MSTRG.25558.1 | Fibroblast growth factor receptor 2-like | G | — | G | — |
| adi_MSTRG.8066.1 | ate_MSTRG.9763.1 | Fibroblast growth factor receptor 3-like | — | S | — | G |
| adi_MSTRG.8065.1 | ate_MSTRG.9740.1 | Fibroblast growth factor receptor 3-like | — | S | PC | G |
| adi_MSTRG.8057.1 | ate_MSTRG.9728.1 | Fibroblast growth factor receptor 3-like | — | S | PC | G |
| adi_MSTRG.9430.1 | ate_MSTRG.3763.1 | Fibroblast growth factor receptor 4-like | PC | S | PC | — |
| adi_MSTRG.8054.1 | ate_MSTRG.25730.1 | Fibroblast growth factor receptor-like 1 | G | G | PC | — |
| adi_MSTRG.961.1 | ate_MSTRG.7414.1 | Forkhead box protein A2-A-like | — | S | PC | G |
| adi_MSTRG.4321.1 | ate_MSTRG.1275.1 | Forkhead box protein C1-B-like | — | S | PC | G |
| adi_MSTRG.25166.1 | ate_MSTRG.1423.1 | Forkhead box protein C2-B-like | G | G | PC | — |
| adi_MSTRG.9655.1 | ate_MSTRG.17746.1 | Forkhead box protein G1-like | — | S | G | — |
| adi_MSTRG.12519.1 | ate_MSTRG.17857.1 | Forkhead box protein J1-B-like | G | S | G | — |
| adi_MSTRG.16804.1 | ate_MSTRG.1614.1 | Forkhead box protein J3-like | — | S | G | S |
| adi_MSTRG.12764.1 | ate_MSTRG.14741.1 | Frizzled-5-like | G | S | G | G |
| adi_MSTRG.13445.1 | ate_MSTRG.13276.1 | GS homeobox 1-like | PC | S | G | G |
| adi_MSTRG.21417.1 | ate_MSTRG.2816.1 | Homeobox protein Dlx4a-like | G | — | G | S |
| adi_MSTRG.21145.1 | ate_MSTRG.9255.1 | Homeobox protein GBX-2-like | — | G | PC | — |
| adi_MSTRG.9345.1 | ate_MSTRG.9661.1 | Homeobox protein Hox-A4-like | PC | S | G | G |
| adi_MSTRG.21137.1 | ate_MSTRG.9253.1 | Homeobox protein koza-like | G | G | PC | S |
| adi_MSTRG.19273.1 | ate_MSTRG.13715.1 | Homeobox protein Nkx-2.5-like | — | S | PC | G |
| adi_MSTRG.26073.1 | ate_MSTRG.7296.1 | Homeobox protein OTX-like | G | — | PC | — |
| adi_MSTRG.18244.1 | ate_MSTRG.19329.1 | Homeobox protein six1b-like | G | G | PC | S |
| adi_MSTRG.18255.1 | ate_MSTRG.19398.1 | Homeobox protein SIX3-like | G | S | G | G |
| adi_MSTRG.17569.1 | ate_MSTRG.4081.1 | Homeobox protein slou-like | PC | S | G | — |
| adi_MSTRG.22159.1 | ate_MSTRG.16160.1 | Homeobox protein SMOX-3-like | G | G | — | S |

| Transcript ID | | Description | A. digitifera | | A. tenuis | |
| --- | --- | --- | --- | --- | --- | --- |
| A. digitifera | A. tenuis | | PC versus G | G versus S | PC versus G | G versus S |
| adi_MSTRG.11668.1 | ate_MSTRG.12315.1 | Homeobox protein XENK-2-like | G | — | PC | — |
| adi_MSTRG.11663.1 | ate_MSTRG.12330.1 | Homeobox protein XENK-2-like | G | — | PC | — |
| adi_MSTRG.611.3 | ate_MSTRG.22280.1 | LIM domain transcription factor LMO4.2-like | PC | S | — | G |
| adi_MSTRG.8553.1 | ate_MSTRG.8832.1 | LIM/homeobox protein Lhx1-like | — | S | G | G |
| adi_MSTRG.22763.2 | ate_MSTRG.7655.1 | Microphthalmia-associated transcription factor-like | — | G | PC | G |
| adi_MSTRG.1598.1 | ate_MSTRG.883.1 | Neurogenic locus notch homolog protein 1-like | G | G | G | G |
| adi_MSTRG.21759.1 | ate_MSTRG.3777.1 | Nuclear transcription factor Y subunit A-9-like | G | — | — | S |
| adi_MSTRG.2218.1 | ate_MSTRG.18693.1 | Paired box protein Pax-6-like | G | — | PC | — |
| adi_MSTRG.13373.1 | ate_MSTRG.21708.1 | Paired mesoderm homeobox protein 2B-like | G | G | — | G |
| adi_MSTRG.26490.1 | ate_MSTRG.18630.1 | Pancreas transcription factor 1 subunit alpha-like | — | S | G | G |
| adi_MSTRG.15324.1 | ate_MSTRG.18341.1 | Pituitary homeobox 2-like | G | — | G | — |
| adi_MSTRG.10819.1 | ate_MSTRG.19187.1 | POU domain, class 3, transcription factor 3-like | G | G | PC | — |
| adi_MSTRG.11198.1 | ate_MSTRG.25046.1 | POU domain, class 4, transcription factor 3-like | — | S | G | — |
| adi_MSTRG.5374.1 | ate_MSTRG.23610.1 | PR domain zinc finger protein 14-like | G | G | G | — |
| adi_MSTRG.1619.1 | ate_MSTRG.860.1 | Prickle-like protein 3 | PC | S | — | G |
| adi_MSTRG.3836.1 | ate_MSTRG.19931.1 | Protein jagged-1-like | G | — | PC | — |
| adi_MSTRG.16444.1 | ate_MSTRG.20642.1 | Protein SOX-15-like | — | S | G | G |
| adi_MSTRG.12811.1 | ate_MSTRG.26020.1 | Protein Wnt-1-like | — | S | PC | — |
| adi_MSTRG.15556.1 | ate_MSTRG.7658.1 | Protein Wnt-2b-like | G | S | G | — |
| adi_MSTRG.10027.1 | ate_MSTRG.8271.1 | Protein Wnt-7b-like | — | S | G | G |
| adi_MSTRG.3517.1 | ate_MSTRG.11874.1 | Protein Wnt-7b-like | PC | S | — | G |
| adi_MSTRG.1867.1 | ate_MSTRG.15696.1 | Putative transcription factor SOX-14 | G | — | G | — |
| adi_MSTRG.13379.1 | ate_MSTRG.21689.1 | Retinal homeobox protein Rx1-like | — | S | G | G |
| adi_MSTRG.16543.1 | ate_MSTRG.11602.1 | Retinal homeobox protein Rx1-like | — | S | G | G |
| adi_MSTRG.21405.1 | ate_MSTRG.12728.1 | Runt-related transcription factor 3-like | PC | — | PC | — |
| adi_MSTRG.1615.1 | ate_MSTRG.7820.1 | Secreted frizzled-related protein 3-like | — | G | PC | — |
| adi_MSTRG.38.1 | ate_MSTRG.239.1 | T-box transcription factor TBX1-like | — | S | PC | — |
| adi_MSTRG.1605.1 | ate_MSTRG.22318.1 | T-box transcription factor TBX20-like | PC | — | PC | — |
| adi_MSTRG.12532.1 | ate_MSTRG.405.1 | T-box transcription factor TBX20-like | G | S | G | G |
| adi_MSTRG.12667.1 | ate_MSTRG.430.1 | T-box transcription factor TBX3-like | — | G | G | — |
| adi_MSTRG.7351.1 | ate_MSTRG.6146.1 | Tiggy-winkle hedgehog protein-like | — | G | PC | G |
| adi_MSTRG.25198.1 | ate_MSTRG.1447.1 | Transcription factor 25-like | — | S | G | — |
| adi_MSTRG.398.1 | ate_MSTRG.26268.1 | Transcription factor A, mitochondrial-like | — | G | — | S |
| adi_MSTRG.18722.1 | ate_MSTRG.9434.1 | Transcription factor COE2-like | — | G | G | — |
| adi_MSTRG.19828.1 | ate_MSTRG.14515.1 | Transcription factor HES-1-like | G | G | — | S |
| adi_MSTRG.19864.1 | ate_MSTRG.14525.1 | Transcription factor HES-1-like | G | — | PC | S |
| adi_MSTRG.19836.1 | ate_MSTRG.14526.1 | Transcription factor HES-1-like | G | — | PC | S |
| adi_MSTRG.18072.1 | ate_MSTRG.21184.1 | Transcription factor MafF-like | G | S | PC | — |
| adi_MSTRG.19244.1 | ate_MSTRG.25955.1 | Transcription factor Sox-11-like | G | — | G | S |
| adi_MSTRG.16290.1 | ate_MSTRG.11063.1 | Transcription factor SOX-14-like | G | G | G | — |
| adi_MSTRG.14803.1 | ate_MSTRG.7006.1 | Transcription factor Sox-14-like | G | G | G | — |

**Table 3. Continued**

| Transcript ID | | Description | A. digitifera | | A. tenuis | |
| A. digitifera | A. tenuis | | PC versus G | G versus S | PC versus G | G versus S |
|---|---|---|---|---|---|---|
| adi_MSTRG.6401.1 | ate_MSTRG.21433.1 | Transcription factor SOX-14-like | — | S | — | S |
| adi_MSTRG.17475.1 | ate_MSTRG.16150.1 | Transcription factor Sox-21-A-like | — | S | PC | G |
| adi_MSTRG.3082.1 | ate_MSTRG.7083.1 | Transcription factor Sox-2-like | G | — | G | — |
| adi_MSTRG.16590.1 | ate_MSTRG.11585.1 | Transcription factor Sp9-like | G | G | — | S |
| adi_MSTRG.4624.1 | ate_MSTRG.19219.1 | Transcription factor Sp9-like | PC | S | G | — |
| adi_MSTRG.13179.1 | ate_MSTRG.8653.1 | Zinc finger protein GLIS2-like | PC | — | — | S |

320 in P2, and 1,781 in P3 (Fig S2A). These results suggest that *A. digitifera* exhibits a larger number of duplicate genes that are not differentially regulated between transitions. In contrast, *A. tenuis* recruits a higher proportion of duplicated genes with coordinated regulation during early developmental stages, which may reflect a more robust GRN. Furthermore, although 88% (PC versus G) and 80% (G versus S) of the P1 paralogous pairs in *A. digitifera* showed similar expression patterns, 99% (PC versus G) and 97% (G versus S) of the P1 paralogous pairs in *A. tenuis* did so (Fig S2B). Despite this, we observed a concentration of P1 and P2 pairs in the G stage for *A. digitifera* and in the PC stage for *A. tenuis* (Fig S2C and D), suggesting differences in the timing and functional deployment of gene duplicates between the two species.

Functional enrichment analysis of *A. digitifera* P1 paralogs in the PC-versus-G transition revealed an overrepresentation in PC of BPs such as killing cells of another organism (GO:0031640), epigenetic regulation of gene expression (GO:0040029), and gastrulation (GO:0007369). In contrast, ribosome biogenesis (GO:0042254), mitotic cell cycle (GO:0000278), and Wnt/PCP signaling pathway (GO:0060071) were overrepresented in G. Likewise, in *A. tenuis*, positive regulation of IL-8 production (GO:0032757), regulation of cyclin-dependent protein kinase activity involved in G2/M (GO:0010971), and Wnt/PCP signaling pathway (GO:0060071) were overrepresented in PC, and protein processing (GO:0016485), photoprotection (GO:0010117), and immune response (GO:0006955) were overrepresented in G. On the other hand, in *A. digitifera* G-versus-S transition ncRNA processing (GO:0034470), negative regulation of gene expression, epigenetic (GO:0045814), and p38MAPK cascade (GO:0038066) were overrepresented in G, and skeletal muscle thin filament assembly (GO:0030240), migration in host (GO:0044001), and cilium movement (GO:0003341) were overrepresented in S. In *A. tenuis*, skeletal muscle thin filament assembly (GO:0030240) and migration in host (GO:0044001) were overrepresented in G, and epigenetic regulation of gene expression (GO:0040029) was overrepresented in S (Table S13). In other words, during gastrulation, both species recruit differentially expressed paralogous genes (P1) involved in key processes such as epigenetic regulation, signaling, and immune response. However, this recruitment occurs in a species- and stage-specific manner, indicating that duplicated genes may play important regulatory roles during critical stages of embryonic development, and that their expression may be subject to adaptive functional diversification.

In-paralogs were reclassified into three categories (Pk1, Pk2, and Pk3) based on K-means clustering to characterize their coexpression profiles further: Pk1: both in-paralogs were assigned to a cluster; Pk2: only one gene was assigned to a cluster; and Pk3: neither was assigned to a cluster. Overall, 3,611, 4,559, and 2,752 paralogous pairs belonged to Pk1, Pk2, and Pk3 for *A. digitifera*, and 1,584, 415, and 221 pairs belonged to Pk1, Pk2, and Pk3 for *A. tenuis*, respectively (Fig S3A). Expression patterns of the Pk1 paralogous pairs showed greater conservation in *A. tenuis* compared with *A. digitifera*, as ~83% of the *A. tenuis* paralogous pairs showed similar expression patterns. In contrast, in *A. digitifera*, only 47% of the paralogous pairs exhibited similar expression patterns (Fig S3B). Likewise, we found that ~80% of the paralogs in *A. tenuis* exhibit a high correlation (>0.9), whereas 31% do so in *A. digitifera* (Table S14). This suggests a more conserved regulation of expression among duplicates during gastrulation in *A. tenuis*. Most Pk1 and Pk2 paralogs in *A. tenuis* clustered in C1 (Fig S3C and D), whereas in *A. digitifera*, Pk1 mainly clustered in C2 and C5 (Fig S3C) and Pk2 clustered primarily in C1 and C2 (Fig S3D).

Pk1 in-paralogs were enriched in BPs involved in small GTPase-mediated signal transduction (GO:0007264) in AdC1, chromatin organization (GO:0006325) in AdC2, RNA methylation (GO:0001510) in AdC3, positive regulation of canonical Wnt receptor signaling pathway (GO:0090263) in AdC4, ciliary or flagellar motility (GO:0001539) in AdC5, actin-myosin filament sliding (GO:0033275) in AdC6, and establishment of planar polarity of embryonic epithelium (GO:0042249) in AtC1 (Table S15). Pk1 paralogous pairs with different expression patterns in *A. digitifera* were enriched for DNA replication–dependent chromatin assembly (GO:0006335), p38MAPK cascade (GO:0038066), regulatory ncRNA-mediated gene silencing (GO:0031047), and regulation of anatomical structure size (GO:0090066), among others. In *A. tenuis*, the Pk1 paralogs were enriched for epigenetic regulation of gene expression (GO:0040029), developmental growth involved in morphogenesis (GO:0060560), regulatory ncRNA-mediated posttranscriptional gene silencing (GO:0035194), and ectoderm development (GO:0007398) (Table S16). Moreover, Pk1 paralogous pairs present in the clusters up-regulated in G (C2, C3, and C4) were enriched with molecules with roles in the Wnt signaling pathway (GO:0030111), chromatin silencing (GO:0031507), neuron maturation (GO:0042551), and the MAPK cascade (GO:0000165) in *A. digitifera*, and mesenchyme migration (GO:0090131) and chromatin remodeling (GO:0006338) in *A. tenuis* (Table S17).

**Table 4.  Conserved gene coexpression during gastrulation of *A. digitifera* and *A. tenuis*.**

| Clusters | Orthologous pairs | | Description |
| --- | --- | --- | --- |
| | ID (*A. digitifera*) | ID (*A. tenuis*) | |
| AdC1-AtC1 | adi_MSTRG.21405.1 | ate_MSTRG.12728.1 | **Runt-related transcription factor 3-like**[a] |
| | adi_MSTRG.8890.1 | ate_MSTRG.20159.1 | Spondin-1-like |
| | adi_MSTRG.19064.1 | ate_MSTRG.22273.1 | **Protein DVR-1 homolog** |
| | adi_MSTRG.13201.1 | ate_MSTRG.8623.1 | GRB2-associated-binding protein 1-like |
| | adi_MSTRG.26952.1 | ate_MSTRG.9285.1 | **Mothers against decapentaplegic homolog 4-like** |
| | adi_MSTRG.21148.1 | ate_MSTRG.9278.1 | **barH-like 1 homeobox protein** |
| AdC2-AtC1 | adi_MSTRG.1605.1 | ate_MSTRG.22318.1 | T-box transcription factor TBX20-like |
| | adi_MSTRG.7351.1 | ate_MSTRG.6146.1 | Tiggy-winkle hedgehog protein-like |
| | adi_MSTRG.1615.1 | ate_MSTRG.7820.1 | Secreted frizzled-related protein 3-like |
| AdC2-AtC2 | adi_MSTRG.25711.2 | ate_MSTRG.26313.1 | **Bone morphogenetic protein 1-like** |
| AdC3-AtC4 | adi_MSTRG.386.1 | ate_MSTRG.10325.1 | **Doublesex- and mab-3–related transcription factor 3-like** |
| | adi_MSTRG.16290.1 | ate_MSTRG.11063.1 | **Transcription factor SOX-14-like** |
| | adi_MSTRG.16544.1 | ate_MSTRG.11573.1 | **Aristaless-related homeobox protein-like** |
| | adi_MSTRG.5374.1 | ate_MSTRG.23610.1 | **PR domain zinc finger protein 14-like** |
| | adi_MSTRG.14803.1 | ate_MSTRG.7006.1 | **Transcription factor Sox-14-like** |
| AdC4-AtC1 | adi_MSTRG.11668.1 | ate_MSTRG.12315.1 | **Homeobox protein XENK-2-like** |
| | adi_MSTRG.11663.1 | ate_MSTRG.12330.1 | **Homeobox protein XENK-2-like** |
| | adi_MSTRG.3836.1 | ate_MSTRG.19931.1 | **Protein jagged-1-like** |
| | adi_MSTRG.8054.1 | ate_MSTRG.25730.1 | **Fibroblast growth factor receptor-like 1** |
| | adi_MSTRG.26073.1 | ate_MSTRG.7296.1 | **Homeobox protein OTX-like** |
| AdC4-AtC5 | adi_MSTRG.19836.1 | ate_MSTRG.14526.1 | **Transcription factor HES-1-like** |
| | adi_MSTRG.19244.1 | ate_MSTRG.25955.1 | **Transcription factor Sox-11-like** |
| | adi_MSTRG.21417.1 | ate_MSTRG.2816.1 | **Homeobox protein Dlx4a-like** |
| AdC4-AtC4 | adi_MSTRG.1867.1 | ate_MSTRG.15696.1 | **Putative transcription factor SOX-14** |
| | adi_MSTRG.12519.1 | ate_MSTRG.17857.1 | **Forkhead box protein J1-B-like** |
| | adi_MSTRG.13695.1 | ate_MSTRG.18015.1 | **Doublesex- and mab-3–related transcription factor A2-like** |
| | adi_MSTRG.15324.1 | ate_MSTRG.18341.1 | **Pituitary homeobox 2-like** |
| | adi_MSTRG.12532.1 | ate_MSTRG.405.1 | **T-box transcription factor TBX20-like** |
| | adi_MSTRG.3082.1 | ate_MSTRG.7083.1 | **Transcription factor Sox-2-like** |
| AdC5-AtC1 | adi_MSTRG.19273.1 | ate_MSTRG.13715.1 | **Homeobox protein Nkx-2.5-like** |
| | adi_MSTRG.17475.1 | ate_MSTRG.16150.1 | **Transcription factor Sox-21-A-like** |
| | adi_MSTRG.18072.1 | ate_MSTRG.21184.1 | **Transcription factor MafF-like** |
| | adi_MSTRG.12811.1 | ate_MSTRG.26020.1 | **Protein Wnt-1-like** |
| | adi_MSTRG.8071.1 | ate_MSTRG.5924.1 | **Fibroblast growth factor receptor 2-like** |
| | adi_MSTRG.961.1 | ate_MSTRG.7414.1 | **Forkhead box protein A2-A-like** |
| | adi_MSTRG.8075.1 | ate_MSTRG.9709.1 | **Fibroblast growth factor 18-like** |
| | adi_MSTRG.8057.1 | ate_MSTRG.9728.1 | **Fibroblast growth factor receptor 3-like** |
| | adi_MSTRG.8065.1 | ate_MSTRG.9740.1 | **Fibroblast growth factor receptor 3-like** |
| | adi_MSTRG.8066.1 | ate_MSTRG.9763.1 | **Fibroblast growth factor receptor 3-like** |
| AdC5-AtC2 | adi_MSTRG.4321.1 | ate_MSTRG.1275.1 | **Forkhead box protein C1-B-like** |
| | adi_MSTRG.13445.1 | ate_MSTRG.13276.1 | **GS homeobox 1-like** |
| | adi_MSTRG.25899.1 | ate_MSTRG.19738.1 | **Tolloid-like protein 1** |

**Table 4.  Continued**

| Clusters | Orthologous pairs | | Description |
| --- | --- | --- | --- |
| | ID (*A. digitifera*) | ID (*A. tenuis*) | |
| AdC5-AtC3 | adi_MSTRG.12764.1 | ate_MSTRG.14741.1 | **Frizzled-5-like** |
| | adi_MSTRG.13379.1 | ate_MSTRG.21689.1 | **Retinal homeobox protein Rx1-like** |
| | adi_MSTRG.10027.1 | ate_MSTRG.8271.1 | **Protein Wnt-7b-like** |
| | adi_MSTRG.8553.1 | ate_MSTRG.8832.1 | **LIM/homeobox protein Lhx1-like** |
| AdC5-AtC4 | adi_MSTRG.25198.1 | ate_MSTRG.1447.1 | Transcription factor 25-like |
| | adi_MSTRG.9655.1 | ate_MSTRG.17746.1 | **Forkhead box protein G1-like** |
| | adi_MSTRG.18255.1 | ate_MSTRG.19398.1 | **Homeobox protein SIX3-like** |
| | adi_MSTRG.16444.1 | ate_MSTRG.20642.1 | **Protein SOX-15-like** |
| | adi_MSTRG.11198.1 | ate_MSTRG.25046.1 | **POU domain, class 4, transcription factor 3-like** |
| | adi_MSTRG.15556.1 | ate_MSTRG.7658.1 | **Protein Wnt-2b-like** |
| AdC5-AtC5 | adi_MSTRG.17330.2 | ate_MSTRG.15266.1 | **Chordin-like** |
| | adi_MSTRG.6401.1 | ate_MSTRG.21433.1 | **Transcription factor SOX-14-like** |

[a]Bold entries indicate transcripts mentioned in the main text.

Overall, these results indicate that gene duplication events contribute differently to the transcriptional programs governing gastrulation in each *Acropora* species. The higher number of differentially expressed paralogs and the divergent regulation between gene pairs across the evaluated stages in *A. digitifera* suggest a greater regulatory complexity, possibly linked to species-specific innovations or adaptations. In contrast, the more synchronized expression profiles observed in *A. tenuis* paralogs point to a more conserved regulatory program. This divergence highlights how paralogous dynamics and the differential enrichment of biological processes may reflect evolutionary and developmental differences between closely related species, despite sharing morphologically similar embryonic stages.

### Duplication and divergence of components of developmental signaling pathways: rewiring of coexpression networks during *Acropora* gastrulation

Interestingly, Pk1 paralogous pairs included molecules with roles in development and diverse types of TFs. We identify components of the Wnt, Notch, FGF, and BMP pathways in both species, as well as transcription factors (TFs) such as *Otp-like*, *Isx-like*, *Pax-3-like*, *Pax-6-like*, and *HES-1-like* (Table S18). In general terms, *A. digitifera* paralogous pairs showed distinct expression profiles; for example, divergent expression patterns were observed during gastrulation in the pairs annotated as an "intestine-specific homeobox-like" (adi_MSTRG.17500.1 and adi_MSTRG.22167.1) and "Wnt-1-like" (adi_MSTRG.12811.1 and adi_MSTRG.13202.1). Despite differences in expression patterns, the *Isx-like* orthologs showed high sequence similarity (92.5% identical sites), unlike the *Wnt1-like* orthologs (53.4% identical sites). In contrast, the *A. tenuis* paralogs tended to have similar expression patterns, with minor variations in expression levels, as was the case for the pairs annotated as "frizzled-1-like" (ate_MSTRG.12175.1 and ate_MSTRG.12185.1) and "paired box protein Pax-3-B-like" (ate_MSTRG.18681.1 and ate_MSTRG.18693.1). This pattern was also independent of sequence similarity (Fig 4A). These findings reinforce the idea that *A. digitifera* has developed greater

regulatory flexibility between duplicated genes, possibly associated with functional specialization. Moreover, the low sequence similarity observed in some orthologs, such as *Wnt1-like* (53.4%), suggests a greater evolutionary distance that may have facilitated this functional divergence. In contrast, *A. tenuis* exhibits more synchronized transcriptional control between duplicates, which may reflect evolutionary pressure to preserve key regulatory functions during development, which may serve to maintain developmental stability.

To understand how gene duplication and differences in temporal expression shape coexpression networks, we focused on a group of DEGs annotated as "Wnt-1-like protein," as species-specific duplication of Wnt ligands likely reflects the divergence of Wnt signaling between species during early gastrulation. Although adi_MSTRG.12811.1 and adi_MSTRG.13202.1 were identified as paralogs in *A. digitifera*, ate_MSTRG.26020.1 was recognized as a one-to-many ortholog to both *A. digitifera* genes (Fig 4B). All transcripts showed distinct expression patterns (Fig 4C). We obtained three subnetworks (one for each query gene) with marked differences in coexpression partners (Fig 4D). Although some subnetworks shared homologous genes, the direction of correlation with their query gene did not coincide, for example, the orthologs adi_MSTRG.3250.1 and ate_MSTRG.2704.1, annotated as "brachyury protein-like," and adi_MSTRG.15556.1 and ate_MSTRG.7658.1, annotated as "protein Wnt-2b-like." In both cases, the *A. tenuis* genes showed a negative correlation with their *Wnt-1-like* query genes, whereas their orthologous counterparts in *A. digitifera* showed a positive correlation (Fig 4D; Table S19). These results suggest the deployment of distinct coexpression networks because of gene duplication and changes in expression patterns during gastrulation in *Acropora*.

### Divergent regulation of HES-1–like isoform suggests the usage of species-specific transcriptional networks during neurogenesis in *Acropora*

Next, we assessed the distribution of isoforms across species in our dataset. In both species, most genes with AS exhibited two

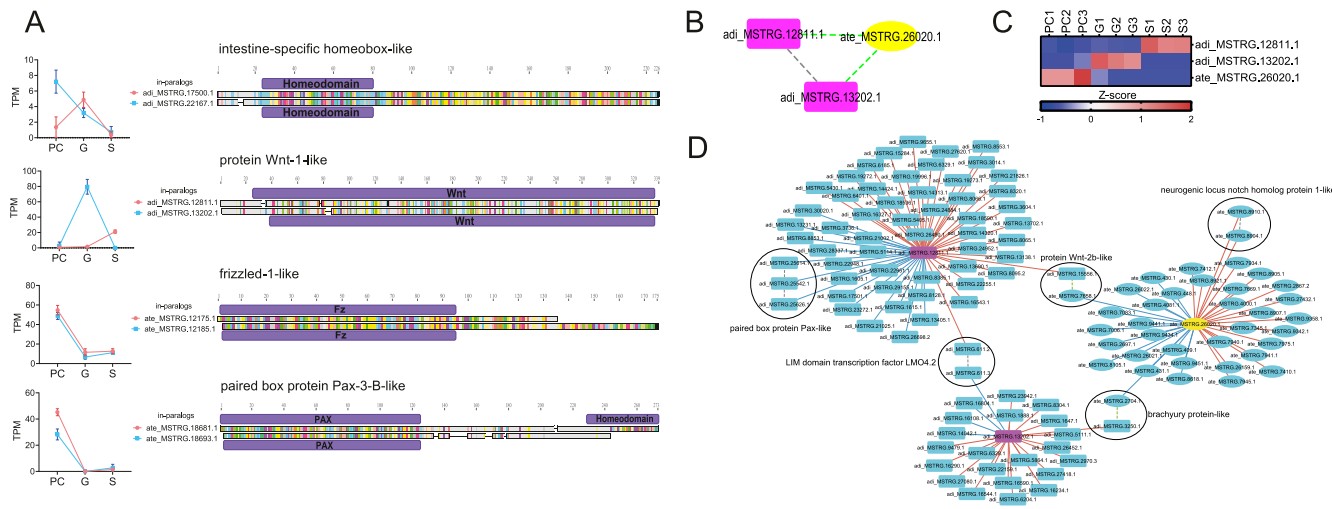

**Figure 4. In-paralogous expression patterns and coexpression networks.**
**(A)** Expression pattern and alignment of *Isx-like*, *Wnt1-like*, *Fz1-like*, and *Pax3b-like* in-paralogous pairs. The y-axis shows the normalized count values in TPM, and the x-axis shows the developmental stages. The alignments highlight nonconserved amino acid residues and domains (Pfams) in light gray and purple, respectively. **(B)** Homology relationships of annotated "*Wnt1-like*" genes. Dashed borders represent homologous relationships, green for orthologs and gray for paralogs. **(C)** Expression patterns of "*Wnt1-like*" annotated genes. **(D)** Correlation networks of *Wnt1-like* homologs. Continuous edges represent positive (red) or negative (blue) correlations and connect genes exhibiting a PCC greater than 0.8. The blue nodes represent the correlated genes, whereas the rectangular purple nodes represent the *A. digitifera* query genes, and the elliptical yellow node represents the query *A. tenuis* gene. Circles around two nodes indicate groups of homologs. Dashed borders represent homologous relationships, green for orthologs and gray for paralogs.

isoforms (69.1% for *A. digitifera* and 93.1% for *A. tenuis*). However, we observed more genes with isoforms in *A. digitifera* (4,706) than in *A. tenuis* (493) and identified important groups of isoforms in both species, such as adi_MSTRG.11010 (RNA-directed DNA polymerase from mobile element jockey-like) with 22 isoforms and adi_MSTRG.13006 (zinc finger protein OZF-like) with seven isoforms for *A. digitifera*, and ate_MSTRG.8994 (tubulin beta-4B chain) with five isoforms and ate_MSTRG.5153 (homeobox protein Meis1-like) with three isoforms for *A. tenuis* (Table S20; Fig S4A and B). We focused our attention on two isoforms annotated as "transcription factor HES-1-like": ate_MSTRG.14525.1 and ate_MSTRG.14525.2 in *A. tenuis*, and its ortholog adi_MSTRG.19864.1 in *A. digitifera*. We selected this molecule because HES genes play a role in the establishment of the nervous system, a process that, according to our results, shows temporal correlation between species, suggesting that neuron specification in both species may be achieved via slightly distinct signaling pathways at the same time. We first evaluated expression patterns and amino acid residue alignments between transcripts (Fig 5A and B). Both ate_MSTRG.14525.1 and ate_MSTRG.14525.2 were mainly expressed in PC, whereas adi_MSTRG.19864.1 was expressed in G and S (Fig 5A). Furthermore, we found differences between isoforms in length and amino acid residue composition (Fig 5B).

To understand the role of isoforms in the changes of transcriptional networks during gastrulation, we compared the correlation interactions between the two *Hes1-like* isoforms in *A. tenuis* and its ortholog in *A. digitifera* (Fig 5C). We found a set of 32 transcripts that were highly correlated in the same direction with both isoforms, which included transcripts annotated as *Sox14-like*, *Sox15-like*, *Chrdl-like*, *Six3-like*, *Hes2-like*, *Ctnnd2-like*, *Wnt7b-like*, and *Bra-like*. We also found

six groups of shared orthologs between one or both isoforms and their orthologous gene in *A. digitifera* that included *Sox2-like*, *Spry-like*, *Kctd6-like*, *Fgfr2-like*, *Barhl1-like*, and *Btbd6-like*.

Despite this, in some cases, the direction of the correlation between orthologous groups and the query genes was not always the same. For example, adi_MSTRG.19864.1 was positively correlated with *Sox2-like* (adi_MSTRG.3082.1) in *A. digitifera*, whereas ate_MSTRG.14525.1 and ate_MSTRG.14525.2 were negatively correlated with *Sox2-like* (ate_MSTRG.7083.1) in *A. tenuis*. In addition, distinct sets of transcripts were uniquely correlated with each isoform. For instance, ate_MSTRG.14525.1 was specifically associated with *Bmp2a-like*, *Six1b-like*, and *Meis1-like* transcripts, whereas ate_MSTRG.14525.2 showed correlations with *Otx-like*, *Jag-like*, and *Gbx2-like*. In the same way, we found genes correlated only with adi_MSTRG.19864 as is the case of *Foxn3-like*, *Sp4-like*, *Ctnnb-like*, and *Gata3-like*. On the other hand, we found that ate_MSTRG.14525.2 presented a positive correlation with components of the NOTCH signaling pathway (*Notch1-like* and *Jag-like*), whereas ate_MSTRG.14525.1 was negatively correlated with this type of gene.

Our results indicate that AS in *Acropora* is a mechanism capable of increasing transcriptomics complexity and fine-tuning development. Similar to paralogous genes, isoform diversity is likely to contribute to generating diversity in developmental programs, thereby fueling adaptation processes in response to environmental and evolutionary pressures. Although gene duplication is expected to create biological novelty via neofunctionalization (Conant & Wolfe, 2008; Birchler & Yang, 2022), AS might function as a modulator of gene expression underlying phenotypic plasticity. More research is necessary to test these ideas.

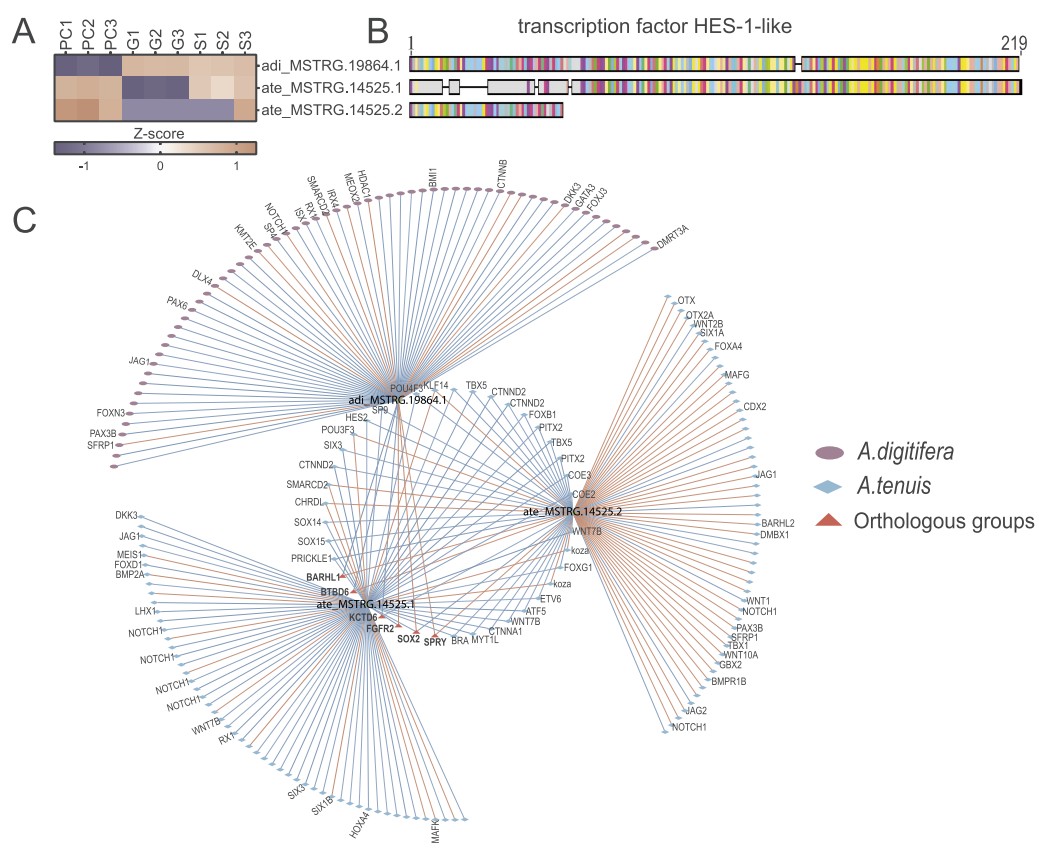

**Figure 5. HES1-like isoform expression patterns and correlation networks.**
**(A, B)** Expression patterns (A) and sequence alignment (B) of *HES1-like* isoforms and their ortholog. **(C)** Correlation networks of *Hes1-like* bait genes. Red and blue borders indicate positive and negative correlation, respectively. Red triangles represent groups of orthologs (gene families) correlated between species. Genes shared between query genes were clustered in the inner circle of interactions, whereas unique interactions were located outside the networks.

# Discussion

### Transcriptional divergence during gastrulation in *Acropora*: revisiting the hourglass model and the search for a phylotypic cnidarian stage

Although gastrulation is an animal symplesiomorphy (Nakanishi et al, 2014), the different ways this process occurs in cnidarians and bilateral organisms exhibit remarkable intra- and intergroup variation (Kraus & Markov, 2017). In other words, there is no strong correlation between gastrulation mode and taxa, as closely related species often exhibit very different gastrulation mechanisms. Still, despite the apparent differences in the morphogenetic processes that give rise to gastrulation, there is a degree of conservation in the underlying molecular mechanisms, suggesting that changes from one mode of gastrulation to another may be due to minor modifications in GRNs (Hayward et al, 2015; Technau, 2020). For example, several signaling pathways (e.g., WNT, FGF, and BMP) and cell movements are conserved across different animal groups. Still, their components are used in various contexts to specify patterns that lead to morphogenetic movements and germ layer formation (Solnica-Krezel, 2005; Technau, 2020). On the other hand, other aspects that affect gastrulation, such as developmental speed, embryo size and shape, yolk amount, blastomere number, or

blastocoel formation (Kraus & Markov, 2017; Schauer & Heisenberg, 2021), are highly variable at the genus level, suggesting the existence of species-specific signaling pathways.

Numerous models have been proposed to predict conservation patterns during animal development; the "hourglass" is the most widely accepted (Irie & Kuratani, 2011, 2014; Kalinka & Tomancak, 2012). This model predicts early and late phases of divergence during ontogeny within a phylum, linked by a morphologically conserved period of mid-embryonic development known as the phylotypic period (Kalinka & Tomancak, 2012). Our results are consistent with this idea, revealing divergent transcriptional programs regulating gastrulation in *Acropora*. These observations suggest that although biological variation in early animal development can lead to diversification of early developmental strategies, the morphological outcome is ultimately subjected to various physical, mechanical, and geometrical constraints (Edelman et al, 2016; Fiuza & Lemaire, 2021; Goodwin & Nelson, 2021). Based on our findings, in *Acropora*, the diversification of GRNs is achieved through the rewiring of ancestral GRNs via the asynchronous expression of distinct yet conserved gene modules, gene duplication, and species-specific isoform expression.

According to the hourglass model, one would expect to observe lower conservation of GRNs during early embryogenesis and

higher conservation during the phylotypic period (Martinez-Morales, 2016). This idea is well supported in bilaterians but remains debated in cnidarians because of the remarkable variability in early embryogenesis and the diverse life cycles (El-Bawab, 2020). Despite this, our results support that GRNs deployed during gastrulation are diverse and may explain the diversification of developmental strategies observed in basal marine invertebrates (Peter & Davidson, 2011; Steventon et al, 2021). In cnidarians, planula larvae are considered the phylotypic stage, representing the most conserved stage in the entire phylum (Kamm et al, 2006). Comparative transcriptomics studies during development in cnidarians are necessary to characterize developmental GRN diversity further and validate the planula as the Cnidarian phylotypic stage.

### Morphological convergence via regulatory divergence: evidence of developmental system drift during gastrulation in *Acropora*

Although the lower sequencing depth for *A. tenuis* and the use of short-read sequencing technologies may have limited the detection of low-abundance transcripts, our analysis reveals species-specific gene expression programs underlying gastrulation in *Acropora*. Despite displaying morphologically similar embryonic stages, *A. digitifera* and *A. tenuis* gastrulating embryos exhibit species-specific, divergent transcriptional profiles during this morphological transition, indicating that morphologically similar coral developmental stages arose from distinct transcriptional programs. Although stage-specific transcriptional profiles indicated that *A. tenuis* gastrulas closely resembled the early larval stage (S), *A. digitifera* gastrulas were closer to blastulas (PC) (Fig 1B). These results are consistent with the concept of developmental system drift (DSD), which posits that morphologically similar-looking stages or outcomes can be achieved through divergent GRNs (McColgan & DiFrisco, 2024). PCA clustering and DEG distribution show substantial divergence in gene regulatory activity during gastrulation, supporting the idea of DSD during gastrulation in *Acropora*. Our results provide, for the first time, evidence of DSD in corals.

*A. digitifera* had 18,497 DEGs, nearly double the number of DEGs found in *A. tenuis* (9,486), with distinct transitions showing peak transcriptional activity and differential enrichment of biological processes (Fig 2). Furthermore, DEGs were more abundant during the G-versus-S transition in *A. digitifera* and in PC-versus-G transition in *A. tenuis* (Fig S1A). Likewise, we found low overlap in coexpression clusters, with a small proportion (13%) of one-to-one orthologs sharing similar coexpression patterns (Fig S1D and E), as well as differential usage of components of developmental signaling pathways (e.g., Sox, Tbx, Fox, Wnt, BMP) and transcription factors (Table S1). As mentioned before, our results support that similar morphologies (developmental stages) can arise or be maintained through distinct and divergent transcriptional programs, indicating that developmental phenotypic homology does not necessarily imply GRN conservation (True & Haag, 2001; McColgan & DiFrisco, 2024). Broadly, our results support the idea that developmental changes during animal evolution will likely result from regulatory divergence via modifications in cis-regulatory elements (Gaunt & Paul, 2012) rather than genetic novelty. Further research on basal invertebrate developmental

GRNs is necessary to test this idea. Although future studies employing long-read or single-cell approaches may enhance resolution, our dataset and analysis provide a solid foundation for understanding the regulatory divergence underlying gastrulation in *Acropora*.

### Rewiring the network: orthologous genes can follow different routes to function

Our analysis revealed that *A. digitifera* and *A. tenuis* shared 10,308 orthologous gene pairs, representing 6,896 orthologous groups, likely originating from an ancestral *Acropora* genome (Shinzato et al, 2021). Although 86.5% (5,965) accounted for "one-to-one" relationships, only ~27% (1,629) were DEGs in both species, revealing a role of this subset of molecules during gastrulation (Fig 3A and B; Table S3). Furthermore, among the 1,629 differentially expressed orthologous pairs, only ~13% (215) exhibited conserved expression patterns across species. Most orthologous DEGs exhibited asynchronous expression, species-specific stage associations, and differential integration into distinct transcriptional modules showing divergent regulatory partners and biological enrichments (Fig 3B, D, and E). These findings support the idea that gene conservation does not necessarily reflect regulatory or temporal conservation, as orthologous genes often follow divergent regulatory trajectories (Sobral et al, 2009; Gharib & Robinson-Rechavi, 2011). Our results revealed a flexible use of conserved molecules with roles in development, as shown by the divergent expression of genes such as *Wnt2b-like*, *SP9-like*, and *HES-1-like* (Table S3). Our findings support the idea that shared developmental molecules can be reprogrammed to generate lineage-specific programs (Davidson & Erwin, 2006) and suggest that regulatory rewiring plays a crucial role in shaping functional divergence during development in *Acropora* (Peter & Davidson, 2011). Our results reflect both the regulatory flexibility and robustness of coral development, as well as their capacity to adapt to environmental challenges transcriptionally. Regulatory program divergence may enhance phenotypic plasticity in corals, while increasing their adaptive potential (Palumbi et al, 2014; Shinzato et al, 2021) as decentralized GRNs facilitate evolutionary innovation without compromising structural integrity (Davidson & Erwin, 2006).

On the other hand, the identification of a small set of 370 orthologs (*Wnt2b-like*, *Sox2-like*, and *Six3-like*), up-regulated at the gastrula stage in both species and enriched in biological processes such as axis specification, neurogenesis, and endoderm formation (Tables S7 and S8), indicates the existence of ancestral GRNs underlying gastrulation in *Acropora*. This core GRN is likely fundamental for establishing the coral larval body plan and, therefore, is under strong purifying selection (Erwin & Davidson, 2009). This result is consistent with the idea of conserved developmental "kernels" that function as backbone networks upon which species-specific regulatory modifications can be layered (Davidson & Erwin, 2006). Moreover, up-regulation of this module in both species at the gastrula stage suggests that these processes are fundamental for morphogenesis, revealing a conserved transcriptional core underlying the gastrulation processes. Module conservation may be explained by intense selective pressures

experienced by developing *Acropora* embryos in their environment, as both species undergo gastrulation as part of their planktonic life before becoming demersal (Ball et al, 2004). *Acropora* embryos become free-swimming larvae within 36–48 h postfertilization (Okubo & Motokawa, 2007; Reyes-Bermudez et al, 2016) and thus rely on a fully functional nervous system to coordinate ciliary-based swimming, spatial orientation, and interaction with the environment (Attenborough et al, 2019; Pysanczyn et al, 2023).

Evidence shows that neuron progenitors first appear during the blastula stage in *Nematostella vectensis* and *Acropora* embryos, with differentiation into functional neurons occurring during gastrulation (Nakanishi et al, 2012; Attenborough et al, 2019). Moreover, the finding of enrichment of retinoic acid (RA) signaling in this core module (Table S8) suggests that this pathway plays an essential role in neurogenesis in Anthozoans, as the pathway is not involved in neuronal differentiation in other cnidarians (Bouzaiene et al, 2007). RA is also a key regulator of morphogenetic movements in vertebrates (Janesick et al, 2018; Gur et al, 2022), which indicates that recruitment to neurogenesis might be an Anthozoan innovation. Likewise, *Nkx2.5*-like and *Foxa2*-like, known regulators of oral/aboral axial specification in *Nematostella* (Wijesena et al, 2017), were among the subset of orthologs showing consistent coexpression patterns in *Acropora* (Table 4). Altogether, our results indicate that despite the divergent expression of a significant proportion of orthologous genes, ecological demands are the primary source of developmental constraint, preserving core regulatory circuits that enhance fitness and ensure survival. Further research is necessary to assess the extent of conservation of this core gastrulation GRN in cnidarians.

## Modularity and plasticity of *Acropora* developmental GRNs: coral resilience under rapid environmental change

Low conservation in expression patterns during gastrulation might be explained by species-specific GRN modules showing divergent temporal expression patterns (Fig S1C and D). GRNs were well-defined coexpression units containing conserved transcription factors (e.g., Sox, Fox, Tbx) and signaling pathway components (e.g., BMP, Wnt) asynchronously expressed between species (Fig S1D, Table 4). For example, in *A. digitifera*, gastrula-up-regulated DEGs could be divided into three well-defined GRNs. Although the first one was also expressed in *A. tenuis's* gastrula, the second one was up-regulated in blastula (PC), and the third one in early larvae (S) (Fig 3D). Likewise, when we compared coexpression clusters, we found that each cluster can be subdivided into well-defined GRNs, mapped to clusters from the other species showing divergent temporal profiles (Fig 3E). These results support the idea that the rewiring of transcriptional networks shapes animal development primarily via changes in regulatory elements rather than by creating new genes (Erwin & Davidson, 2009). The concept of modular GRNs posits that sets of coregulated genes act semi-independently (Alon, 2007), promoting adaptations by allowing subunits to diverge peripherally without disrupting the entire network (Wagner et al, 2007; Peter & Davidson, 2011).

*Acropora's* modular organization, underlying axial symmetry, neuronal differentiation, and cilium movement are good examples of developmental flexibility, as they reveal lineage-specific regulation of key biological processes experiencing intense selective pressure (Fig 2C and D). In particular, neuron differentiation that occurs in both species during the G-to-S transition via nonidentical gene sets (Fig 2C and D) indicates how putative ancestral GRN modules can be incorporated into broader species-specific regulatory contexts, supporting the idea that developmental programs in basal metazoans show both deep conservation and great flexibility (Gilbert et al, 2024). Larval stages of reef-building corals are vital for dispersal and ecological resilience; therefore, the ability to rewire developmental programs might reflect the robustness of coral development and an ancient evolutionary mechanism underlying adaptive radiation while promoting phenotypic stability (Kenkel & Matz, 2016; Hazraty-Kari et al, 2022).

Interestingly, our analysis highlighted the importance of lncRNAs as regulators of species-specific GRNs during *Acropora* gastrulation, as they represented ~10% of DEGs (Fig S1B) and were more diverse in embryonic stages than in early larva in both species (Fig 1D). As lncRNAs have been associated with priming pluripotent cell populations for differentiation (Ghosal et al, 2013), it is reasonable to think that in *Acropora*, lncRNAs are involved in priming embryonic cell populations for differentiation and therefore putative candidates for the initiation of cell lineage–specific differentiation programs (Lu et al, 2021). Finding that the abundance of lncRNAs differed between species at the blastula stage (AdPC: Q1-Q2; AtPC: Q3-Q4) indicates species-specific use for these molecules. More research using in situ hybridization is necessary to test this idea. Because lncRNA's ortholog inference and annotation are based on predictions of folding structure rather than sequence, we were limited to discussing our results based on abundance and distribution.

## Gene duplication: source of regulatory innovation and GRN modularity

Gene duplication has been linked with increased genome regulatory complexity and biological innovations (Wagner, 2008; Gagnon-Arsenault et al, 2013), providing opportunity for subfunctionalization or transcriptional redundancy (Birchler & Yang, 2022). This study identified 10,922 paralogous pairs differentially expressed during gastrulation in *A. digitifera* and 2,220 in *A. tenuis*, suggesting more gene duplication events during independent evolution in this coral species (Shinzato et al, 2021). These results are consistent with previous studies on *A. digitifera*, showing that approximately one-third of predicted genes in *A. digitifera* result from tandem duplications (Noel et al, 2023). Interestingly, expression profiles of DEG paralogous pairs significantly differed between species, which is consistent with the idea that gene duplication has played a pivotal role in the diversification of the genus (Mao & Satoh, 2019; Shinzato et al, 2021). In *A. digitifera*, ~47% of paralogous pairs showed similar temporal expression patterns, contrasting with the ~83% in *A. tenuis* (Table S14). This is consistent with a previous study showing low gene expression correlation between paralogous pairs during *A. digitifera* life cycle (Mao & Satoh, 2019). Moreover, for *A. digitifera*, paralogous DEGs were more abundant at the gastrula, and for *A. tenuis* in the blastula,

indicating distinct temporal regulation during development (Fig S2C and D).

Likewise, divergent in-paralogous DEGs are frequently coexpressed with distinct gene clusters, indicating integration of these molecules into different regulatory programs (Fig 4D; Table S14), and therefore are fundamental for network reprogramming and rewiring (Gagnon-Arsenault et al, 2013). Similarly, the finding of species-specific functional enrichment of paralogous subsets reflected species-specific recruitment of duplicated gene products. Although in *A. digitifera*, paralogous genes are enriched in processes like epigenetic gene regulation, stress response, and MAPK signaling, in *A. tenuis*, duplicated genes were associated with ectoderm development and chromatin remodeling (Tables S13, S15, S16, and S17). Surprisingly, we found the noncanonical Wnt/PCP signaling pathway (GO:0060071) enriched in in-paralogous groups for both species, indicating species-specific diversification of Wnt signaling (Tables S13, S15, S16, and S17). Wnt/PCP coordinates cell polarity in the ectoderm and guides embryo elongation (Momose et al, 2012), and PCP disruption has been shown to modify endoderm-specific gene expression (Lapébie et al, 2014). Therefore, species-specific control of Wnt/PCP signaling during gastrulation might reflect the divergence of this morphogenetic process in *Acropora*.

Altogether, our findings indicate that although *A. digitifera* integrates duplicated genes into divergent developmental GRN modules (e.g., Wnt-1-like, Isx-like) (Fig 4), likely to result in subfunctionalization or neofunctionalization (Gagnon-Arsenault et al, 2013), *A. tenuis* uses its duplicated gene products to generate robustness via regulatory redundancy and stability (Diss et al, 2014). Interestingly, paralogous groups in both species were enriched in processes such as innate immunity and responses to external stimuli, which are fundamental for early development and are likely subjected to strong evolutionary constraints (Voolstra et al, 2011; Palmer et al, 2012). There is evidence showing that in *A. digitifera*, duplicated genes are associated with the expansion of innate immunity (Hamada et al, 2013; Poole & Weis, 2014) and photoprotective enzyme gene families (Kashimoto et al, 2021), as well as with loci with roles in modulating transcriptional network dynamics during thermal stress or acidification (Shinzato et al, 2021). Furthermore, there is evidence that genes under selection during thermal stress frequently map to duplicated loci, harboring adaptive alleles that increase fitness (Noel et al, 2023; Shah et al, 2023). These observations indicate that in *Acropora*, gene duplication plays a crucial role in maintaining developmental flexibility and ecological resilience (Shinzato et al, 2021), by providing both redundancy and the opportunity for subfunctionalization. In other words, paralogous genes give corals a flexible "genomic toolkit" that increases phenotypic plasticity and adaptation potential (Kashimoto et al, 2021; Yoshioka et al, 2023).

### Alternative splicing: species-specific fine-tuning of GRNs

Like gene duplication, alternative splicing (AS) is an essential source of regulatory complexity and flexibility during animal development, as it allows functional diversification without altering gene content (Singh & Ahi, 2022). Our analysis revealed that in both species, most alternative spliced genes possess at least two

isoforms, with *A. digitifera* (4,706) showing a greater number than *A. tenuis* (493) (Table S20). These might be explained by differences in transcriptome coverage between species (Table 1). Despite this, genes experiencing AS and the number of isoforms predicted from each locus were species-specific, with transcript variants often exhibiting divergent expression patterns (Table S20). These results and the observation that transcript variants from a single locus mapped to distinct gene modules in both species indicate that in *Acropora*, AS plays a vital role in rewiring GRNs during gastrulation (Fig 5C). Our findings indicate that similar to gene duplication, AS is a source of diversification of developmental GRN, by generating regulatory divergence (Singh & Ahi, 2022). Although studies on the role of AS in coral development are limited, focusing mainly on responses to stress and symbiosis (Huang et al, 2019; Yu et al, 2021), parallels with vertebrates indicate a role of AS during early animal development (Tian et al, 2020; Liu et al, 2022) and species divergence (Singh et al, 2017; Verta & Jacobs, 2022).

Moreover, our results indicate that AS might be responsible for phenotypic robustness and ecological adaptability in *Acropora*, by allowing species-specific expression of conserved regulatory transcription factors (Fig 5). Phenotypic plasticity and developmental GRNs' robustness are very relevant in corals, as early developmental stages occur in plankton and are therefore susceptible to environmental fluctuations (Reyes-Bermudez et al, 2016). By adjusting gene activity, AS has the capacity to modulate stress responses and facilitate local adaptations during early coral development. Interestingly, finding temporal divergent isoforms for the HES-1–like transcription factor in *A. tenuis* (Fig 5A), whose expression patterns differed from the single *A. digitifera* ortholog, and play roles in neuronal differentiation, indicates that in both species, neurogenesis is achieved at the same time (Fig 2C and D) via slightly different pathways, as isoforms interact with overlapping but nonidentical GRNs and show differential interactions with key regulators of neuronal differentiation such as *Sox2*-like and *Notch1*-like genes (Richards & Rentzsch, 2015) (Fig 5C). These results indicate that DSD also occurs at the cellular level by maintaining distinct and conserved cellular phenotypes but allowing diversification of the underlying differentiation programs. More research integrating single-cell and long-read transcriptomics, spatial expression profiling, and functional genomics is necessary to understand the role of AS in coral developmental resilience in the context of climate change.

Altogether, our results support the DSD concept at different levels, as we demonstrated that both developmental outcomes (stages) and cellular phenotypes (neurons) can originate from divergent and rewired GRNs. Finding an *Acropora* conserved GRN "kernel" perimetrically modified in a species-specific manner revealed corals' use of highly modular and flexible developmental programs. This study not only strengthens the position of *Acropora* as a powerful model for evolutionary developmental biology but also reveals how developmental plasticity may underpin coral resilience in the face of environmental change.

### Final remarks

In this study, we reported for the first time evidence of DSD in corals. Our findings demonstrated that morphologically similar

coral developmental stages arose from distinct transcriptional programs. We identified a conserved "gastrulation genetic toolkit" of 1,629 "one-to-one" orthologs differentially expressed during the developmental progression. These molecules included widespread genes previously associated with early animal development. Despite this, orthologous genes showed significant temporal variation and modular expression divergence, indicating diversification of developmental programs rather than functional conservation. Yet, we identified a subset of 370 DEGs that showed synchronized expression during gastrulation in both species. This conserved module regulates axial specification, endoderm differentiation, and neuronal differentiation via Wnt and RA signaling. This supports the idea that developmental complexity is achieved via lineage-specific peripheral modifications of conserved developmental "kernels." Finding a role of RA signaling during neurogenesis in corals indicates divergence of the regulatory programs underlying neuronal differentiation in Cnidaria, as RA signaling–mediated neuronal differentiation has only been reported for *Nematostella*.

Furthermore, our results revealed that gene duplication and AS are crucial in the divergence and reprogramming of developmental GRNs. Although *A. digitifera* might use gene duplication to favor functional diversification via neofunctionalization, *A. tenuis* might promote GRNs' stability and robustness. Interestingly, in both species, in-paralogous DEGs were enriched with molecules involved in environmental interactions and immune responses, suggesting an adaptive role of these molecules during early coral development. Likewise, duplicated genes were enriched in components of the noncanonical Wnt/PCP signaling pathway, indicating species-specific divergence of Wnt signaling during gastrulation.

# Materials and Methods

## Ethics statement

Ethical approval was not required for collecting or maintaining biological samples.

## Biological material collection and transcriptome description

The early stages of development of the reef-building corals *A. digitifera* and *A. tenuis* were collected at the Sesoko Island Research Station, Okinawa, Japan, during the annual coral spawning event (June–July) in 2012 under collection permits issued by the local authority to the University of Ryukyus. The methods used for fertilization, embryo maintenance, RNA extraction, library construction, and sequencing are described in Reyes-Bermudez et al (2016). Briefly, gametes from six colonies were mixed in separate containers for 2 h until the first cleavage occurred. Developing embryos were maintained in filtered seawater (1 *µ*m) at 26°C until they reached the desired developmental stage. Total RNA was extracted from key developmental stages corresponding to blastula (PC) (12 h postfertilization, HPF), gastrula (G) (24 HPF), and postgastrula (S) (48 HPF) (Fig 1A).

Sequencing libraries were generated from total RNA using the Nextera library preparation kit from Illumina. Libraries were sequenced on the Illumina GAIIx platform in paired-end 50-bp mode. For both species, each stage was represented by three biological replicates containing 1,000 embryos each. Our dataset consisted of nine libraries representing three stages (3X each) (PC, G, and S) from two *Acropora* species (*A. digitifera* and *A. tenuis*). In this study, we report new RNA-seq data for *A. tenuis*, available in the NCBI SRA database under accession ID PRJNA1118343. For *A. digitifera*, we used raw RNA-seq data previously published by Reyes-Bermudez et al (2016), obtained under the same experimental conditions. These data were downloaded from the DNA Data Bank of Japan (DDBJ) under BioProject ID PRJDB3244.

## Data processing and transcriptome assembly

Read quality for each library was evaluated using FastQC (Andrews, 2010). Then, adapters and low-quality reads were eliminated with Trimmomatic (Bolger et al, 2014) using the following parameters: ILLUMINACLIP: NexteraPE-PE.fa:2:30:10 HEADCROP:16 MINLEN:34. The NexteraPE-PE.fa file containing the adapter sequences was used for this step. The reads were aligned to their respective reference genomes (Shinzato et al, 2021) using HISAT2 (Kim et al, 2019) with the --very-sensitive option. Genetic models for both species are available in the OIST Marine Genomics Unit database (https://marinegenomics.oist.jp).

StringTie (Pertea et al, 2015) was used to generate transcriptome assemblies (including isoforms) of each replicate using the corresponding species' genome as a reference (Shinzato et al, 2021). The reference genomes used in this study were downloaded from GenBank for *A. digitifera* (GenBank assembly accession: GCA_014634065.1) and *A. tenuis* (GenBank assembly accession: GCA_014633955.1). The "merge" function (without providing a GTF reference annotation file) generated nonredundant merged transcripts per stage. Finally, the assembled transcripts (for each stage) were extracted in FASTA format from the GTF file using the utility GffRead (Pertea & Pertea, 2020). A FASTA file with the reference genomic sequences was provided for this operation. The -e option of StringTie was used, along with the reference annotation file, to estimate the expression levels of the transcripts. Then, the Python script prepDE.py provided by StringTie was used to obtain the read count matrix (Pertea et al, 2015).

## Annotations and functional enrichment analysis

Initially, the transcriptomes were annotated according to the predicted proteome for *A. digitifera* and *A. tenuis* (Shinzato et al, 2021) using BLASTx with the Blast+ 2.2.31 package, using an *E*-value limit of $1 \times 10^{-5}$. Transcripts not mapped to the proteomes were aligned against the NCBI nonredundant protein sequence database (ftp.ncbi.nlm.nih.gov/blast/db/). We used Sma3s (Muñoz-Mérida et al, 2014) to extract annotations from the UniProtKB/SWISS-PROT database, which included associated GO terms and metabolic pathway descriptions. Transcripts not annotated using either of these approaches were evaluated in terms of their coding potential using CPC2 (Kang et al, 2017). Alternatively, TransDecoder

(https://transdecoder.github.io/) was used to identify putative coding sequences using BLASTp against reference proteomes and HMMER v3.1 (Eddy, 2011) against the Pfam protein domain database (Finn et al, 2014). GO-term functional enrichment analysis was performed using BinGO (Maere et al, 2005). The hypergeometric test with FDR correction was employed, with a significance level (*P*-adjustment) set at 0.05. A custom file containing the GO terms associated with each transcript was created to perform this analysis.

### Differential expression analysis

Differential expression analyses were performed using the DESeq2 package (Love et al, 2014) in the statistical environment R. Expression was compared between successive pairwise developmental stages, resulting in two comparisons for each species: (1) PC versus G and (2) G versus S. DEGs were defined as those with a | log$_2$FC| > 1, with Benjamini–Hochberg FDR-adjusted *P*-values ≤ 0.05. A variance-stabilizing transformation (VST) was applied to the counts to perform the clustering of the samples by PCA and K-means. The web tool iDEP 1.1 (Ge et al, 2018) was used to perform K-means clustering. The number of clusters was defined as six using the elbow method, and the maximum number of iterations was 1,000.

### Homologous expression analysis

OrthoMCL (Li et al, 2003) was used to identify homologous transcripts between *Acropora* species using the peptides predicted by TransDecoder as input. Clusters of orthologs and in-paralogs ("recent" paralogs) were identified. The all-v-all BLAST step was set with the parameter (–F″m S″) for the best detection of orthologs (Moreno-Hagelsieb & Latimer, 2008). Paralogous gene pairs were classified according to their differential expression patterns during early *Acropora* development into three categories: P1, paralogs in which both genes were differentially expressed; P2, paralogous pairs in which only one of the genes was differentially expressed; and P3, paralogous pairs in which neither gene was differentially expressed. Likewise, the paralogs were reclassified into three categories Pk1, Pk2, and Pk3, according to their expression patterns previously defined by K-means clustering: Pk1, both genes were assigned to a cluster; Pk2, only one gene was assigned to a cluster; and Pk3, neither gene was assigned to a cluster.

### Sequence alignments

Alignments between homolog and isoform sequences were performed using MUSCLE alignment tools (Edgar, 2022) from Geneious 2024.0.5 (https://www.geneious.com) with default parameters. The InterProScan plugin for Geneious (https://www.geneious.com/plugins/interpro-scan) was used to annotate proteins with protein families and domains automatically.

### Construction of correlation networks

Correlation networks were constructed using the CoExpNetViz (Tzfadia et al, 2016) plugin of Cytoscape (Shannon et al, 2003) based on differential expression data. Pearson's correlation coefficient (PCC) was calculated as a similarity measure for gene expression measurements. A series of genes of interest (query or bait genes) in both *A. digitifera* and *A. tenuis* were used as input, along with gene expression data from both species. Cutoff thresholds of 0.5 for the lower percentile and 0.95 for the upper percentile were set to represent only the most significant correlations at the network's edges (Tzfadia et al, 2016). Then, genes correlated with query genes were grouped into gene families according to their homology relationships (orthology or paralogy).

## Data Availability

The raw RNA-seq data generated in this study for *A. tenuis* have been deposited in the NCBI Sequence Read Archive (SRA) under BioProject accession number PRJNA1118343.

## Supplementary Information

## Acknowledgements

We would like to extend our sincere gratitude to all members of the Sesoko Marine Station at the University of the Ryukyus in Okinawa, as well as Dr. Miriam Brandt, for their invaluable assistance during the coral spawning process. We also thank Hidaka (Ryukyus) and Mikheyev (OIST) laboratory members for field assistance and technical support, respectively. We thank the Marine Biological Laboratory of the University of Chicago for the opportunity to attend the 2025 "*Gene Regulatory Networks for Development*" advanced course, which provided a very stimulating environment to learn and discuss different aspects of developmental GRNs. The OIST DNA sequencing section in Onna, Okinawa, performed sequencing. This work was supported by a postdoctoral fellowship awarded to A Reyes-Bermúdez from the Japanese Society for the Promotion of Science (2010-2012) and internal funds from the Okinawa Institute of Science and Technology Graduate University awarded to the Mikheyev laboratory in 2012.

### Author Contributions

JP Ossa-Gómez: conceptualization, resources, data curation, software, formal analysis, supervision, funding acquisition, validation, investigation, visualization, methodology, project administration, and writing—original draft, review, and editing.

HA Rodríguez-Cabal: conceptualization, resources, data curation, software, formal analysis, supervision, funding acquisition, validation, investigation, visualization, methodology, project administration, and writing—original draft, review, and editing.

A Reyes-Bermúdez: conceptualization, resources, data curation, software, formal analysis, supervision, funding acquisition, validation, investigation, visualization, methodology, project administration, and writing—original draft, review, and editing.

# Life Science Alliance

**Conflict of Interest Statement**

The authors declare that they have no conflict of interest.

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
