## [Reviewer comments · Life Science Alliance]

Life Science Alliance

Developmental system drift and modular gene regulatory networks shape gastrulation in *Acropora*.

Juan Ossa-Gomez, Hector Rodriguez-Cabal, and Alejandro Reyes-Bermudez

DOI: <https://doi.org/10.26508/lsa.202503293>

Corresponding author(s): Alejandro Reyes-Bermudez, University of the Amazon

Review Timeline:

Submission Date:	2025-03-04
Editorial Decision:	2025-04-15
Revision Received:	2025-06-10
Editorial Decision:	2025-07-17
Revision Received:	2025-07-30
Editorial Decision:	2025-07-31
Revision Received:	2025-08-01
Accepted:	2025-08-04

Scientific Editor: Sarita Hebbar

Transaction Report:

April 15, 2025

Re: Life Science Alliance manuscript #LSA-2025-03293-T

Alejandro Reyes-Bermudez
Universidad de la Amazonia

Dear Dr. Reyes-Bermudez,

Thank you for submitting your manuscript entitled "Differential expression of taxonomically restricted in-paralogs and isoforms results in the use of divergent transcriptional networks during gastrulation in two *Acropora* species." to Life Science Alliance. The manuscript was assessed by two expert reviewers, whose comments are appended to this letter.

Overall, both the reviewers indicate that the findings of this manuscript are of potential value. In line with the reviewers' evaluation, we invite you to submit a revised manuscript addressing the reviewers' comments.

We ask that you consider all of the reviewers' suggestions to edit and revise the manuscript. Specifically, the revised version must

-contextualise the results (reviewer 2, paragraphs 2-4)

-acknowledge the weaknesses and limitations as indicated by the reviewers (reviewer 2, paragraph 5) including the sensitivity of the method used here and observed discrepancy with previous publication (Reviewer 1)

Thank you for this interesting contribution to Life Science Alliance. We are looking forward to receiving your revised manuscript.

Sincerely,

Sarita Hebbar, PhD
Scientific Editor
Life Science Alliance
<http://www.lsjournal.org>

B. MANUSCRIPT ORGANIZATION AND FORMATTING:

Reviewer #1 (Comments to the Authors (Required)):

Cnidarians exhibit different mode of gastrulation. To gain insights of gene regulatory network involved in the divergent modes of gastrulation, this study compared gene expression profiles at blastula, gastrula and post gastrula stages between two phylogenetically distant species of the genus *Acropora*, *A. digitifera* and *A. tenuis*, which was estimated to diverge approximately 50 million years ago. The authors identified 1629 orthologous DEGs shared between species with distinct expression profiles during *Acropora* gastrulation. These molecules were associated with neuronal differentiation, axis formation, and germ layer differentiation, mapping to WNT, BMP, NOTCH, and retinoic acid signaling pathways. In addition, they identified species-specific transcriptional networks based on the differential expression of in-paralogs and species-restricted isoforms during gastrulation. For both species, in-paralog groups were enriched with molecules with roles in immune responses, interactions with the environment, and the non-canonical Wnt/PCP pathway. They insist that species-specific diversification of transcriptional networks during gastrulation in *Acropora*.

The informatics analyses of RNA-seq data were carried properly, and the interpretation of the results seems reasonable. Since the data set shown here provides basic information for future studies about this evo-devo issues in cnidarians, I recommend publication of this manuscript in LSA after the authors would provide an answer to a comment below.

Lines 186-187: Aligned reads were assembled, resulting in 38,110 merged transcripts for *A. digitifera* and 28,284 for *A. tenuis* (Table 1). According to Shinzato et al. (2020), the genome size and predicted gene number in *A. digitifera* were 419 Mb and 26,060 and in *A. tenuis* 403 Mb and 22,802, respectively. I wonder why the number of merged transcripts differed so much between *A. digitifera* (38,110) and *A. tenuis* (28,284), nearly 10,000 transcripts difference. This was mainly caused by "Libraries were sequenced on the Illumina GAIIx platform in paired-end 50 bp mode", namely short reads assembly. (Line 687-688). The PacBio Iso-Seq platform looks more desirable method. It is highly likely that this difference affected results obtained by following analyses. Therefore, the authors should comment this point after the description of this result (line 187).

Minor

Line 144: "AS" should be changed to "As".

Line 152-153: "50 million years ago". Please add reference(s).

Reviewer #2 (Comments to the Authors (Required)):

In this paper, the authors have used RNAseq to compare important developmental transitions between two divergent corals. This is important because very similar stages of development can seemingly have very different molecular controls underpinning them. This work has successfully identified key gene expression differences between the blastula, gastrula, and sphere stages of development in *A. digitifera* and *A. tenuis*, including orthologs, paralogs, and isoforms that may contribute to these differences. The authors demonstrate that gastrulation profiles differ between the two species, both in the timing of when different gene expression occurs and the types of functional groupings of the DEGs. This work also highlights conserved developmental modules, enhancing understanding of the similarities of development across eumetazoans.

While I believe the authors have made important and informative findings, I do have some concerns with the organization of the paper, in particular in the presentation of the results. Overall, the paper suffers from a lack of contextualization of the results. Limited attempts are made to guide the reader through the thought process of the authors, or even to explain what conclusions the authors are drawing from their findings. I think this could be improved by using conclusion statements to the paragraphs and major sections. Without these, there is no indication of the significance of the results presented in any given portion of the paper. It might even make sense for this manuscript to combine the results and discussion sections.

For example, there are many results related to developmental pathways and system development, but the authors make no mention of the significance of these differences to the ways that *A. digitifera* and *A. tenuis* develop or survive in their environments.

Similarly, the significance of gene orthologs (beginning at line 248) and in-paralogs (beginning at line 328) would benefit from better introductions, whether those take place in the introduction or in the results sections. When the authors reach conclusions statements, such as those at line 339/340, they are missing necessary context to showcase the significance of these results. It might especially help to compare and contrast what the findings of the orthologs and in-paralogs sections mean biologically.

Finally, the authors do acknowledge this weakness in the discussion, but the results would be strengthened by molecular (such as in situ or IHC) and morphological (such as imaging to examine differences in ciliation, see line 533) data to support the differences in gene expression and gene ontology enrichment. This is likely too extensive to request in the context of this study, but, in its absence, I think there should be a bit more discussion of why these were limitations that could not be met and which types of studies would be the best to follow up this work based on current findings.

Minor concerns and corrections:

There is extensive discussion of the hourglass model in the introduction, but I'm not sure this provides useful background to the paper as it is currently structured (paragraphs beginning at lines 91 and 106). Instead, I think it might make sense to use the information from the beginning of the discussion (lines 457 - 481) as important introductory material and move the material about the hourglass model fully to the discussion.

In the figures, move the letters to the upper left.

Line 225 - Define "BPs". If this is the gene ontology "Biological Processes", is there any reason why Cellular Component and Molecular Function terms were not included in these results?

Line 241 and Fig. 2 - Should group C6 have associated GO terms listed?

Line 421 - What does organization of isoforms "in pairs" refer to? The in-paralogs?

Line 404 - It is not clear why Wnt-1-like was selected out of the various genes to examine in a network.

Line 427 (Fig 5A and B) - It is not clear why HES-1-like has been selected for examination of isoforms.

Line 475 - Spelling error in abbreviation of GRN

Line 499 - What module is being referred to as a "conserved module" here?

Line 527 and 551 - References are made to "heterochronic genes," but I don't believe this term applies. The authors have discussed differences in timing but do not have any evidence of specific genes that control the timing, which would be the "heterochronic genes."

10/06/2025

To: Dr. Sarita Hebbar. Scientific Editor. LSA.

Re: Manuscript ID: LSA-2025-03293-T.

Former title: "Differential expression of taxonomically restricted in-paralogs and isoforms results in the use of divergent transcriptional networks during gastrulation in two *Acropora* species."

New title: "Developmental system drift and modular gene regulatory networks shape gastrulation in *Acropora*".

Running title: "DSD shape gastrulation in *Acropora*".

Authors: Juan P. Ossa-Gómez, Héctor A. Rodríguez-Cabal, Alejandro Reyes-Bermúdez.

Dear Editor,

Thank you for the referees' reports and for encouraging us to revise and resubmit the manuscript previously entitled "*Differential expression of taxonomically restricted in-paralogs and isoforms results in the use of divergent transcriptional networks during gastrulation in two Acropora species.*" The manuscript has been revised in response to the referee's suggestions and uploaded to the LSA server for further consideration. Below, we respond (normal font) point-for-point to the referee's comments (bold font). After revision, we think the title "*Developmental system drift and modular gene regulatory networks shape gastrulation in Acropora*" better describes our findings.

We considered that the current version of our manuscript significantly improved following the reviewers' comments and suggested changes.

Sincerely,

Alejandro Reyes-Bermúdez.

On behalf of the authors.

Reviewer #1 (Comments to the Authors (Required)):

Mayor:

1. Lines 186-187: Aligned reads were assembled, resulting in 38,110 merged transcripts for *A. digitifera* and 28,284 for *A. tenuis* (Table 1). According to Shinzato et al. (2020), the genome size and predicted gene number in *A. digitifera* were 419 Mb and 26,060 and in *A. tenuis* 403 Mb and 22,802, respectively. I wonder why the number of merged transcripts differed so much between *A. digitifera* (38,110) and *A. tenuis* (28,284), nearly 10,000 transcripts difference. This was mainly caused by "Libraries were sequenced on the Illumina GAIIx platform in paired-end 50 bp mode", namely short reads assembly. (Line 687-688).

46 This is a very interesting point and one of the main difficulties associated with comparative
47 transcriptomics. Short read length (50 bp, paired-end) can indeed hinder accurate assembly and
48 contribute to increased transcriptome fragmentation; however, we believe that the main reason for
49 the difference in the number of assembled transcripts between species is due to sequencing depth.
50 *A. digitifera* had approximately 30.5 million reads compared to 22.9 million in *A. tenuis*,
51 facilitating the detection of more low-abundance transcripts in *A. digitifera*, thereby increasing the
52 total number of assembled transcripts reported. Although we missed the resolution for comparing
53 low-abundant transcripts, we were still able to compare a large proportion of differentially
54 expressed genes (DEGs) during gastrulation across species. These observations were added to the
55 methods section.

56

57 **2. The PacBio Iso-Seq platform looks more desirable method. It is highly likely that this**
58 **difference affected results obtained by following analyses. Therefore, the authors should**
59 **comment this point after the description of this result (line 187).**

60

61 We thank the reviewer for their observation and fully agree that long-read sequencing platforms,
62 such as PacBio Iso-Seq, represent a technically superior alternative, as they enable the capture of
63 full-length transcripts, thereby reducing assembly fragmentation and improving isoform
64 identification. However, our data was generated in 2012, when this technology was not yet widely
65 used for RNA sequencing. These observations were added to the methods section.

66

67 **Minor:**

68

69 **1. Line 144: "AS" should be changed to "As".**

70

71 Thank you for the observation, however, "AS" is capitalized because it refers to the acronym for
72 "Alternative Splicing", so it should not be changed to "As". This is presented in the abbreviation
73 section.

74

75 **2. Line 152-153: "50 million years ago". Please add reference(s).**

76

77 Thank you for the suggestion. A reference for the statement "50 million years ago" has now been
78 added to the document.

79

80 **Reviewer #2:**

81

82 **Mayor:**

83

84 **1. While I believe the authors have made important and informative findings, I do have**
85 **some concerns with the organization of the paper, in particular in the presentation of the**
86 **results. Overall, the paper suffers from a lack of contextualization of the results. Limited**
87 **attempts are made to guide the reader through the thought process of the authors, or even**
88 **to explain what conclusions the authors are drawing from their findings. I think this could**
89 **be improved by using conclusion statements to the paragraphs and major sections. Without**
90 **these, there is no indication of the significance of the results presented in any given portion**
91 **of the paper. It might even make sense for this manuscript to combine the results and**

92 **discussion sections.**

93

94 We agree with the comment and thank the reviewer for this observation. We have edited the
95 result section, including conclusive statements in major sections that help to contextualize our
96 findings. The edited version will enable the reader to follow and better understand our ideas and
97 the biological significance of our results. We think that, it is not necessary to unify the results and
98 discussion sections, as the use of conclusion statements in both sections makes it easy to
99 understand our results and the statements that follow.

100

101 **4. For example, there are many results related to developmental pathways and system**
102 **development, but the authors make no mention of the significance of these differences to**
103 **the ways that *A. digitifera* and *A. tenuis* develop or survive in their environments.**

104

105 We agree with the comment and thank the reviewer for this observation. We have edited the
106 result and discussion sections, including statements that link the significance of changes in gene
107 expression (between species) of developmental orthologous genes and the environment.

108

109 **5. Similarly, better introductions, whether in the introduction or in the results sections,**
110 **would benefit the significance of gene orthologs (beginning at line 248) and in-paralogs**
111 **(beginning at line 328).**

112

113 We agree with the comment and thank the reviewer for this observation. We have edited the
114 result section, including introductory statements regarding the significance of orthologs and in-
115 paralogs.

116

117 **6. When the authors reach conclusions statements, such as those at line 339/340, they are**
118 **missing necessary context to showcase the significance of these results. It might especially**
119 **help to compare and contrast what the findings of the orthologs and in-paralogs sections**
120 **mean biologically.**

121

122 We agree with the comment and thank the reviewer for this observation. We have edited the
123 result section, including statements that explain the biological significance of changes in
124 expression of orthologs “one to one” and in-paralogs between species.

125

126 **7. Finally, the authors do acknowledge this weakness in the discussion, but the results**
127 **would be strengthened by molecular (such as *in situ* or IHC) and morphological (such as**
128 **imaging to examine differences in ciliation, see line 533) data to support the differences in**
129 **gene expression and gene ontology enrichment. This is likely too extensive to request in the**
130 **context of this study, but, in its absence, I think there should be a bit more discussion of**
131 **why these were limitations that could not be met and which types of studies would be the**
132 **best to follow up this work based on current findings.**

133

134 We agree with the comment and thank the reviewer for this observation. We agree that
135 microscopy and comparing the spatial localization of DEGs of interest between species would
136 strengthen the manuscript. Unfortunately, we did not fix embryos for *in situ* hybridization nor
137 conduct microscopy on developing larvae. We collected biological material in 2012 at the

138 Sesoko research station in Japan, which lacked high-end microscopy equipment at the time. We
139 have acknowledged this throughout the document and recommend that future studies
140 characterizing the spatial expression of the differentially expressed genes (DEGs) reported here
141 are necessary to understand the relevance of our findings fully.

142

143

144 **Minor concerns and corrections:**

145

146 **1. There is extensive discussion of the hourglass model in the introduction, but I'm not sure**
147 **this provides useful background to the paper as it is currently structured (paragraphs**
148 **beginning at lines 91 and 106). Instead, I think it might make sense to use the information**
149 **from the beginning of the discussion (lines 457 - 481) as important introductory material**
150 **and move the material about the hourglass model fully to the discussion.**

151

152 We agree with the comment and have extensively edited the manuscript following the
153 suggestion.

154

155 **2. In the figures, move the letters to the upper left.**

156

157 Thank you for the comment. The change to the figures has been made.

158

159 **3. Line 225 - Define "BPs". If this is the gene ontology "Biological Processes", is there any**
160 **reason why Cellular Component and Molecular Function terms were not included in these**
161 **results?**

162

163 Indeed, "BPs" refers to *Biological Processes* terms from the Gene Ontology. This is now stated in
164 the abbreviation section. We chose to focus exclusively on BPs because we consider this category
165 to summarize the function of DEGs, laying down a conceptual framework that can be easily
166 compared across species. Still, we acknowledge the value of including the *Molecular Function*
167 and *Cellular Component* categories; however, we opted to omit them to avoid unnecessary length
168 in an already extensive manuscript. However, we are open to including a complementary analysis
169 in the supplementary material if the reviewer considers it appropriate.

170

171 **4. Line 241 and Fig. 2 - Should group C6 have associated GO terms listed?**

172

173 Thank you for your comment. We did not find any enriched GO terms in group C6, so no
174 associated terms were listed. This is now stated in the text and figure legend.

175

176 **5. Line 421 - What does organization of isoforms "in pairs" refer to? The in-paralogs?**

177

178 Thank you for your question. Organization of isoforms "in pairs" refers to genes showing two
179 alternative isoforms. This term does not refer to in-paralogs. This is better explained in the
180 results section.

181

182 **6. Line 404 - It is not clear why Wnt-1-like was selected out of the various genes to examine**
183 **in a network.**

184
185
186
187
188
189
190
191
192
193
194
195
196
197
198
199
200
201
202
203
204
205
206
207
208
209
210
211
212
213
214
215
216
217
218
219
220
221
222
223
224

We selected Wnt-1 because the Wnt gene family plays a key role in several developmental processes across metazoans, including body axis formation, tissue differentiation, and regeneration. Additionally, Wnt-1-like provided a valuable opportunity to study how species-specific duplication events of developmental genes can lead to the diversification of gene regulatory networks (GRNs). This is now stated in the results section.

7. Line 427 (Fig 5A and B) - It is not clear why HES-1-like has been selected for examination of isoforms.

We selected HES-1-like because it belongs to the HES gene family, which is well known for its crucial role in regulating nervous system development across metazoans, a process that, according to our results, occurs simultaneously in both species. Finding two HES-1 isoforms in *A. tenuis*, whereas only a single isoform was detected in *A. digitifera*, gives us an ideal opportunity to explore how post-transcriptional regulatory variation may lead to developmental GRN diversification in *Acropora*. This is now stated in the result section.

8. Line 475 - Spelling error in abbreviation of GRN

Thank you for pointing out the typo. The abbreviation has been corrected from "GNR" to "GRN".

9. Line 499 - What module is being referred to as a "conserved module" here?

We refer to the conserved module of 1,629 differentially expressed genes whose orthologs were expressed in both species. We included a reference to Figure 3B to clarify the sentence in the discussion section.

10. Line 527 and 551 - References are made to "heterochronic genes," but I don't believe this term applies. The authors have discussed differences in timing but do not have any evidence of specific genes that control the timing, which would be the "heterochronic genes."

We thank the reviewer for the comment and agree that the term "heterochronic genes" does not apply in that context. We used the term "heterochronic genes" to refer to orthologous genes that showed differences in the timing of their expression between species. However, we fully agree that, strictly speaking, the term "heterochronic genes" refers to genes that directly regulate developmental timing. Accordingly, we have edited the manuscript and changed the term "heterochronic genes" to "genes with divergent temporal expression patterns" throughout the document.

July 17, 2025

Re: Life Science Alliance manuscript #LSA-2025-03293-TR

Prof. Alejandro Reyes-Bermudez
University of the Amazon
Parcela 65 El Manantial
Florencia, Caquetá 0000
Colombia

Dear Dr. Reyes-Bermudez,

Thank you for submitting your revised manuscript, entitled "Developmental system drift and modular gene regulatory networks shape gastrulation in Acropora", to Life Science Alliance. The manuscript has been seen by one of the original reviewers whose comments are appended below. While the reviewer continues to be overall positive about the work in terms of its suitability for Life Science Alliance, some important issues remain. These are listed below. Please note all changes in manuscript files must be clearly indicated.

- Minor recommendations made by reviewer 2 for the revised version must be followed.
- As suggested by reviewer 1 in the first evaluation, kindly include a statement in the results section highlighting that the difference in the number of merged transcripts could also arise from the technology used in this work in comparison to other published studies.
- You will need to incorporate any points from the Conclusion section into the Discussion section.
- In the methods section, please provide permit details for specimen collection if applicable.
- Please clarify which samples were collected and processed for RNA extraction and sequencing as part of this study. For those sample(s) please deposit with a public repository and include an accession number in the Data Availability statement.
- Please add a call-out for Figure S4A-B to your main manuscript text

Our general policy is that papers are considered through only one revision cycle; however, given that the suggested changes are relatively minor, we are open to one additional short round of revision. Please note that we will expect to make a final decision without additional reviewer input upon re-submission.

Please submit the final revision within one month, along with a letter that includes a point by point response to the remaining reviewer comments.

To upload the revised version of your manuscript, please log in to your account: <https://lsa.msubmit.net/cgi-bin/main.plex>
You will be guided to complete the submission of your revised manuscript and to fill in all necessary information.

- A letter addressing the reviewers' comments point by point.
- An editable version of the final text (.DOC or .DOCX) is needed for copyediting (no PDFs).
- High-resolution figure, supplementary figure and video files uploaded as individual files: See our detailed guidelines for preparing your production-ready images, <https://www.life-science-alliance.org/authors>
- Summary blurb (enter in submission system): A short text summarizing in a single sentence the study (max. 200 characters including spaces). This text is used in conjunction with the titles of papers, hence should be informative and complementary to the title and running title. It should describe the context and significance of the findings for a general readership; it should be written in the present tense and refer to the work in the third person. Author names should not be mentioned.

B. MANUSCRIPT ORGANIZATION AND FORMATTING:

Sincerely,

Sarita Hebbar, PhD
Scientific Editor
Life Science Alliance
<http://www.lsjournal.org>

Reviewer #2 (Comments to the Authors (Required)):

This manuscript provides interesting insights into differences in developmental patterning in two recently diverged species. The authors show that two species of *Acropora* use slightly different gene clusters, with further divergence provided by differential splicing and genome duplication, at conserved developmental timepoints. Overall, this manuscript is vastly improved in readability and comprehensibility from the initial submission. I only have a few minor recommendations that I think could be improved upon.

Ln 205 - This paragraph notes that the distribution of coding sequences is "slightly more abundant" in certain ranges. Looking at the corresponding figure, I see differences that appear to be around 1%, which seem likely to be non-statistically significant. Could the authors provide more information about what this slight difference means, especially if it is not statistically significant?

Ln 232 - Is there a comparison statement missing in this paragraph? The authors begin by comparing the PC vs. G transition, which appear to have different GO terms enriched in the two species. But then the authors say "similarly" and compare the G vs. S, which also appear to have very different GO terms enriched. I'm not sure what is similar about these, which makes me wonder if a sentence was missed? Or it's possible the paragraph just needs to be rearranged a bit to improve readability.

Ln 257, 353, 399, 417 - These sections would all benefit from conclusion statements that highlight the proposed significance of this section of the results. It would help lead the reader through the results.

1

2 **To:** Dr. Sarita Hebbar. Scientific Editor. LSA.

3

4

Re: Manuscript ID: LSA-2025-03293-T.

5

6

Title: “Developmental system drift and modular gene regulatory networks shape gastrulation in *Acropora*”.

7

8

9

Authors: Juan P. Ossa-Gómez, Héctor A. Rodríguez-Cabal, Alejandro Reyes-Bermúdez.

10

11

Dear Editor,

12

13

Thank you for the referees' reports and for encouraging us to revise and resubmit the manuscript. The manuscript has been revised in response to the referee's suggestions and uploaded to the LSA server for further consideration. Below, we respond point-by-point to the referee's and editors' comments.

16

17

18

We consider that the manuscript has significantly improved because of the revisions and hope it will now meet the standards for publication.

19

20

21

Sincerely,

22

23

Alejandro Reyes-Bermúdez.

24

On behalf of the authors.

25

26

Editorial

27

28

-As suggested by reviewer 1 in the first evaluation, kindly include a statement in the results section highlighting that the difference in the number of merged transcripts could also arise from the technology used in this work in comparison to other published studies.

31

32

A statement highlighting that differences in the number of merged transcripts reported in this study and previous works might be explained by the methods used in our study. (L178-183).

33

34

35

-You will need to incorporate any points from the Conclusion section into the Discussion section.

36

37

38

We have edited the conclusion section, making sure that all points stated in it are also included in the discussion.

39

40

41

-In the methods section, please provide permit details for specimen collection if applicable.

42

43

Details have been provided (L840-843).

44

45

-Please clarify which samples were collected and processed for RNA extraction and

46 sequencing as part of this study. For those sample(s) please deposit with a public repository
47 and include an accession number in the Data Availability statement.

48
49 Clarification has been made (L853-857).

50
51 **-Please add a call-out for Figure S4A-B to your main manuscript text**

52
53 The call out has been added L 494.

54
55 **Reviewer #2:**

56
57 **-Ln 205 - This paragraph notes that the distribution of coding sequences is "slightly more**
58 **abundant" in certain ranges. Looking at the corresponding figure, I see differences that**
59 **appear to be around 1%, which seem likely to be non-statistically significant. Could the**
60 **authors provide more information about what this slight difference means, especially if it is**
61 **not statistically significant?**

62
63 Thank you for this observation. We agree that visual differences between quartiles are relatively
64 small (~1%). To address this, we conducted a chi-squared goodness of fit test to evaluate whether
65 the distribution of coding sequences among quartiles (Q1–Q4) was statistically different from a
66 uniform distribution (i.e., 25% expected per quartile). The results showed significant deviations
67 only in the PC and G stages of *A. digitifera* ($p < 0.05$), while the remaining stages—including *A.*
68 *tenuis*—did not show significant differences ($p > 0.05$). We have clarified this in the revised text
69 to avoid overinterpretation and now highlight that this pattern is more evident in specific stages of
70 *A. digitifera* (L197-212).

71
72 **-Ln 232 - Is there a comparison statement missing in this paragraph? The authors begin by**
73 **comparing the PC vs. G transition, which appear to have different GO terms enriched in the**
74 **two species. But then the authors say "similarly" and compare the G vs. S, which also appear**
75 **to have very different GO terms enriched. I'm not sure what is similar about these, which**
76 **makes me wonder if a sentence was missed? Or it's possible the paragraph just needs to be**
77 **rearranged a bit to improve readability.**

78
79 Thank you for this observation. You are correct that the use of “Similarly” was not appropriate in
80 this context, as we were not indicating a direct similarity between the results of the PC vs G and
81 G vs S transitions, but rather introducing a parallel analysis of a different developmental transition.
82 To avoid confusion and improve clarity, we have revised this section by removing “Similarly” and
83 reorganizing the paragraph to clearly distinguish the two comparisons as separate analytical
84 segments (L235).

85
86 **-Ln 257, 353, 399, 417 - These sections would all benefit from conclusion statements that**
87 **highlight the proposed significance of this section of the results. It would help lead the reader**
88 **through the results.**

89
90 We thank the reviewer for this valuable suggestion. In response, we have revised the manuscript
91 to include summary statements at the end of the indicated sections (L266-269, 283-285, 363-366,

92 388-393, 427-435, 452-458). These statements aim to clarify the biological relevance of the
93 findings and guide the reader through the interpretation of the results. We believe these additions
94 improve the coherence and readability of the manuscript.
95

July 31, 2025

RE: Life Science Alliance Manuscript #LSA-2025-03293-TRR

Prof. Alejandro Reyes-Bermudez
University of the Amazon
Parcela 65 El Manantial
Florencia, Caquetá 0000
Colombia

Dear Dr. Reyes-Bermudez,

Thank you for submitting your revised manuscript entitled "Developmental system drift and modular gene regulatory networks shape gastrulation in Acropora.". We would be happy to publish your paper in Life Science Alliance pending final revisions necessary to meet our formatting guidelines.

- Please add an Author Contributions section to your main manuscript text
- Please add ORCID ID for the corresponding author - you should have received instructions on how to do so
- Please add the X and Bluesky handles of your host institute/organization as well as your own or/and one of the authors in our system
- Please be sure that the authorship listing and order is correct

A. FINAL FILES:

B. MANUSCRIPT ORGANIZATION AND FORMATTING:

**Submission of a paper that does not conform to Life Science Alliance guidelines will delay the acceptance of your

manuscript.**

The license to publish form must be signed before your manuscript can be sent to production. A link to the license to publish form will be available to the corresponding author only. Please take a moment to check your funder requirements.

Sincerely,

Sarita Hebbar, PhD
Scientific Editor
Life Science Alliance
<http://www.lsajournal.org>

August 4, 2025

RE: Life Science Alliance Manuscript #LSA-2025-03293-TRRR

Prof. Alejandro Reyes-Bermudez
University of the Amazon
Parcela 65 El Manantial
Florencia, Caquetá 0000
Colombia

Dear Dr. Reyes-Bermudez,

Thank you for submitting your Research Article entitled "Developmental system drift and modular gene regulatory networks shape gastrulation in *Acropora*". It is a pleasure to let you know that your manuscript is now accepted for publication in Life Science Alliance. Congratulations on this interesting work.

DISTRIBUTION OF MATERIALS:

Again, congratulations on a very nice paper. I hope you found the review process to be constructive and are pleased with how the manuscript was handled editorially. We look forward to future exciting submissions from your lab.

Sincerely,

Sarita Hebbar, PhD
Scientific Editor
Life Science Alliance
<http://www.lsajournal.org>